# Causal language modeling can elicit search and reasoning capabilities on logic puzzles

**Kulin Shah** [*]
UT Austin
kulinshah@utexas.edu

**Nishanth Dikkala**
Google Research
nishanthd@google.com

**Xin Wang**
Google Research
wanxin@google.com

**Rina Panigrahy**
Google Research
rinap@google.com

## Abstract

Causal language modeling using the Transformer architecture has yielded remarkable capabilities in Large Language Models (LLMs) over the last few years. However, the extent to which fundamental search and reasoning capabilities emerged within LLMs remains a topic of ongoing debate. In this work, we study if causal language modeling can learn a complex task such as solving Sudoku puzzles. To solve a Sudoku, the model is first required to search over all empty cells of the puzzle to decide on a cell to fill and then apply an appropriate strategy to fill the decided cell. Sometimes, the application of a strategy only results in thinning down the possible values in a cell rather than concluding the exact value of the cell. In such cases, multiple strategies are applied one after the other to fill a single cell. We observe that Transformer models trained on this synthetic task can indeed learn to solve Sudokus (our model solves $94.21\%$ of the puzzles fully correctly) when trained on a logical sequence of steps taken by a solver. We find that training Transformers with the logical sequence of steps is necessary and without such training, they fail to learn Sudoku. We also extend our analysis to Zebra puzzles (known as Einstein puzzles) and show that the model solves $92.04\%$ of the puzzles fully correctly. In addition, we study the internal representations of the trained Transformer and find that through linear probing, we can decode information about the set of possible values in any given cell from them, pointing to the presence of a strong reasoning engine implicit in the Transformer weights [2].

## 1 Introduction

Language models using the Transformer architecture [VSP+17] have displayed remarkable abilities on a variety of Machine Learning tasks over the last few years [BMR+20, RWC+19]. Trained with simply the task of predicting the next token on huge amounts of text, these models display highly performant and deep language understanding skills. In order to make progress on achieving a human-like artificial intelligence, one of the most important ability is the ability to perform human-like reasoning and planning. Although LLMs have displayed a seemingly remarkable ability to excel at reasoning and planning tasks as well, it is a ongoing debate as to whether this ability comes from a true understanding and reasoning of the underlying problem or some other process which simulates reasoning but can be highly brittle. For instance, although LLMs show remarkable

---

[*]Work done during an internship at Google Research.

[2]The code is available at https://github.com/kulinshah98/llm-reasoning-logic-puzzles

38th Conference on Neural Information Processing Systems (NeurIPS 2024).

| Model | Percent of puzzles solved fully | Percent of cells answered correctly |
|-------|--------------------------------|-------------------------------------|
| GPT-4o | 0% | 9.5% |
| Gemini-1.5 Pro | 0% | 10.2% |

Table 1: Results of 4-shot with CoT prompting on Sudoku solving by two of the frontier LLM models. They solved 0 % of the puzzles completely right and their accuracy on a per cell basis was around 9-10% (close to random guessing).

performance on benchmarks requiring non-trivial reasoning and planning skills such as MATH [HBK⁺21], HumanEval [CTJ⁺21] and others, there is research showing that these abilities can be extremely brittle or worse, the model is simply performing 'approximate retrieval' [VOSK22, DLS⁺24].

In this work, we aim to understand how complex a reasoning task can Transformers trained with next-token prediction solve by focusing on a set of synthetic tasks: logic puzzles. In this work we focus our analysis on two types of logic puzzles: Sudoku puzzles and Zebra puzzles.

- **Sudoku.** In the classic variant of Sudoku, we are given a $9 \times 9$ grid where each cell is to be occupied by a number in the range $\{1, 2, \ldots, 9\}$. The constraints are that the numbers along each row and column should be unique. In addition, the numbers within each $3 \times 3$ mini-grid should also be unique. Given a set of initially filled positions, the goal is to figure out the values that can occur in the unfilled cells. In standard Sudoku puzzles, there will always only exist a unique solution to the puzzle.

- **Zebra Puzzles.** There are a more verbal style of a puzzle (Figure 3) where we need to fill in values in a grid again but this time the type of possible constraints is much richer. These are also known as Einstein riddles.

Focusing on synthetic tasks like this gives a precise handle on what data the model has seen, and allows us to also control the difficulty of reasoning required for the task, see e.g. [LHB⁺23, AZL23, LSL⁺23, LAG⁺22]. Prior works have studied how causal language modeling with Transformers performs on synthetic tasks such as learning how to make valid moves in Othello, learning context-free grammars, learning deterministic finite automata and learning specific algorithmic tasks [LAG⁺22, LHB⁺23, NLW23, AZL23, YXLAZ24a, YXLAZ24b] (See Appendix B for a detailed discussion on related work). Compared to these, Sudoku puzzles present a more challenging task. The Sudoku environment is a highly challenging Constraint Satisfaction Problem (CSP) and determining the value in even a single cell can require highly complex reasoning involving multiple steps. In general the extension of the puzzle to $n \times n$ grids is known to be NP-complete [YS03]. Same is the case for Zebra puzzles. However, we will consider a class of logic puzzles which can be solved in polynomial time. This class still remains non-trivial to learn. For an idea of how challenging these can be, we performed a small experiment on how well some frontier LLM's of today can solve Sudoku puzzles. We prompted them in a 4-shot manner with 4 Sudoku puzzles (we serialize a puzzle by converting it into a sequence of (row, column, value) triplets) and their corresponding solutions given before asking for the solution for a 5th test Sudoku puzzle. We evaluated 3000 examples on Gemini-1.5 Pro and GPT-4o. We observed that neither models are able to solve any of the puzzles fully correctly. The results are summarized in Table 1.

## 1.1 Our setup

We treat each Sudoku/Zebra puzzle as a sequence to sequence problem. In a Sudoku, given the sequence of filled cell positions and their values, the model needs to output the sequence of unfilled cell positions and their corresponding values. Similarly in a Zebra puzzle, we are given all the clues and the possible values for the characteristics in a sequential manner and we need to predict the values in the grid. To focus attention on a model's reasoning abilities, we abstract out symbolic versions of the Zebra puzzles. This means we refer to each person as an entity indexed by a number and their favorite color or car becomes an attribute number. For both types of puzzles, clearly, the order in which the model outputs the sequence of filled cells doesn't matter as long as the values are correct. However, we will see that the order in which the solutions are presented to the model during training makes a significant difference in the final performance of the model.

We consider a dataset of Sudoku puzzles of varying difficulty levels from [Rad20]. In addition, we use a Sudoku solver which employs a set of 7 strategies that humans commonly use for solving Sudokus. Given these set of 7 strategies the solver iteratively scans through all unfilled cells and checks if progress can be made using one or more of the strategies. If it finds a cell where progress can be made if fills in its value and repeats the process of searching for the next cell to fill. Although some of the 7 strategies are simple and direct, some of them are highly non-trivial and non-local. From the dataset, we filter out those puzzles which cannot be solved by our solver and end up with 1.9M examples. This ensures that all our puzzles are solvable in polynomial time [3].

We can characterize the size of a Zebra puzzle by a tuple of two numbers: the number of entities and the number of attributes. Each clue in a puzzle is one of 7 different types. We generate around 320,000 Zebra puzzles of sizes varying between (3,3) to (6,6) in the following manner. We first design a human-like solver for these puzzles which tries to solve the puzzles in an iterative manner without backtracking. This solver runs in time cubic in the number of clues of the puzzle. When generating a puzzle of a certain size, we iteratively keep adding clues to a clue set until our solver is able to solve the puzzle. This way, we can ensure that, similar to our Sudoku puzzles, all our sampled Zebra puzzles are also solvable efficiently. Note that, even in the symbolic format, there are an exponential number of puzzles possible implying that our train set and test set won't overlap with a very high probability.

## 1.2 Our results

In this section, we provide an overview of our results. We mainly focus on the Sudoku puzzles to explain our results and include a brief discussion on the Zebra puzzles.

Our first experiment studies whether a Transformer is capable of solving the Sudoku puzzle in a fixed cell order (from top-left to bottom-right). This would amount to the model knowing what values to fill in each unfilled cell in a single forward pass. We observe that although the model learns to predict the values for some cells in a puzzle (average cell accuracy **58.64%** across all unfilled cells), in general, this leads to poor accuracy of solving the complete puzzle (**7.2%**).

Observe that solving a Sudoku puzzle can be thought of as finding easy-to-decode cells and then finding correct value at such cells. We combine this observation with insights from Chain-of-Thought prompting and use our solver to provide the order to fill cells for a given puzzle. In this setting, we use the cell positions provided by the solver during the decoding (i.e., position hints of easy to decode cells) and calculate how many cell values the model gets right. In other words, given a prefix of a partially solved puzzle, we query the solver to find out the "easiest" cell position to solve next and then, conditioning on this position, query the model for its value. The average cell accuracy only goes up marginally by about 3%.

To exploit the full value of the order given by our solver, we train the model from scratch using the solver order. This allows the model to learn what is a good strategic order in which to fill the cells. Importantly this order is adaptive based on the puzzle. To train it in this manner, we first feed each puzzle to our solver and collect the sequence of cells it fills in order. We use these sequences as our training data which acts as our *Chain-of-Thought* data for the model. This leads to a much stronger model which is able to solve full Sudoku puzzles to an accuracy of **87.18%** (see Section 3.4).

Given this new model, we again try giving position hints during decoding as above and we see the average cell accuracy shoot up to **99.02%**. This indicates the following. The iterative process of solving Sudokus can be broken down into two steps: (1) searching and finding a cell position where we can apply a subset of the strategies, (2) given a cell position, computing the value that needs to be filled in that position. Step (1) is the harder task for a model to learn. We provide examples of Sudoku puzzles where the model makes a mistake in step (1) where step (2) is quite trivial (see Section 4.1 for more details). To make the model more proficient at solving the puzzle without the position hints, we perform a beam search of width 3 or 5 and notice that this suffices to get stronger full puzzle solving accuracies of **91.36%** and **94.21%** respectively (see Section 3.5).

In an environment where the model needs to search over a set of candidates to take as the next step, recent work by [BN24] demonstrated that next-token prediction might be a flawed objective. Another recent work [LSM⁺24] posit including the entire search trace as part of the training Chain-of-Thought

---

[3]Each of the 7 strategies the solver uses can be generalized to a general $n \times n$ grid and they can be applied in $poly(n)$ time

data to help a Transformer learn tasks involving search and planning dynamics. In contrast to these works, we observe that Transformers trained with the next-token prediction objective and without access to the entire search trace *can* learn complex reasoning tasks.

Finally, we further ask if we can see a similarity between the model's way of solving the puzzles to a humans/solvers way of solving the puzzle. We study this via probing which has been a technique to understand the *latent* conceptual content, see e.g. [LHB+23, AZL23, PCV23, NLW23, JRR+24]. In particular, works such as [PCV23, NLW23, JRR+24] argue that often simple functions of the model's activations or weights can extract useful latent information (See Appendix B for more details). We study the following via probing. Generally, humans and algorithmic solvers for Sudoku keep track of a possible set of values for each cell at a given state of the board to make progress on solving the Sudoku puzzle. We see that the model also implicitly keeps track of a candidate set and this candidate set matches with the solver's candidate set (see Section 4.2 for more details).

We perform a similar set of experiments as above on Zebra puzzles and observe qualitatively similar trends giving evidence that our conclusions are not limited to the domain of Sudoku (See Appendix I for more details). In summary, our contributions are

1. We show that causal language modeling with the Transformer architecture is capable of learning to perform search and reasoning in highly non-trivial domains like Sudoku and Zebra puzzles.
2. We present evidence that the right form of training data which decomposes the problem into smaller constituents is crucial. However this data is not required to be too descriptive. In particular, it need not contain search traces similar to those provided in [LSM+24].
3. We perform a probing analysis to show that human-like abstract reasoning concepts such as candidate set of values emerge implicitly within the model's activations.

## 2   Preliminaries and setup

In this section, we provide a brief overview of the logic puzzles we consider and the input/output data format that is fed to the model. More details about Zebra puzzle, dataset, architectures and hyperparameters can be found in Appendix D.

**Sudoku puzzle and solver.** The goal of the sudoku puzzle is to fill out the whole board with numbers 1 to 9 without having duplicates in each row, column, and box (See appendix H for more details). Unless specified otherwise, we will use $(r, c)$ to denote the position of a cell on the board and $v(r, c)$ to denote the value at position $(r, c)$ where $r \in \{1, 2, \ldots, 9\}$ denotes the row number of the cell and $c \in \{1, 2, \ldots, 9\}$ denotes the column number of the cell. Additionally, we use $b(r, c)$ to denote the block number (among one of the nine $3 \times 3$ blocks) of the cell at position $(r, c)$. To solve a Sudoku puzzle, a sudoku-solver (and humans up to an extent) keeps track of the candidate set for each of the empty cells. See more details about Candidate set in Appendix H.

As mentioned earlier, the generalized version of Sudoku with board size $n \times n$ is NP-complete [YS03]. This implies that for some Sudoku puzzles, progress likely can not be made using any strategy that executes in polynomial time. We avoid such puzzles by restricting our focus to those Sudoku puzzles that can be solved using a set of 7 well-known and commonly used strategies which are executable efficiently[4]. Further details about each of the strategies is provided in Appendix C. An important point to note is that not all the strategies fill a value in a cell. In fact, only 2 out of 7 strategies that we use, fill a value in a cell and the other strategies are used to eliminate possible values of a cell and narrow down the candidate set at a particular cell. Additionally, some strategies (e.g., XY wing, Unique rectangle) involve reasoning on multiple cells in different rows/columns/blocks and these strategies don't fill a value at any cell and therefore, these strategies need to be applied in combination with other strategies to deduce a value at a cell. Additionally, we only provide the solution list of cell values to the puzzle during training, therefore the model is not getting any direct signal about the strategies that eliminate possible values of a cell and is only getting a signal in combinations of the strategies that deduce a value.

**Dataset, model architecture and training.** Our training dataset for the Sudoku experiment contains 1.8M puzzles and the test dataset contains 0.1M puzzles. Each puzzle also comes with a difficulty

---

[4]The list of the strategies we consider is Lone Single, Hidden Single, Naked Pair, Naked Triplet, Locked Candidate, XY Wing, Unique Rectangle,

rating calculated as follows. To rate a puzzle, a backtracking based solver (different from the one we use to generate our solver-order data) is employed. This solver tries to iteratively make progress on a puzzle using some elimination techniques. When it gets stuck, it makes guesses and tries to solve the puzzle. The difficulty rating is the maximum depth of the guess stack the solver had to use to solve the puzzle. Therefore, even a puzzle rated 0.5 can require complex strategies beyond simple scanning to solve them without guessing. We train a sequence-to-sequence model that takes in as input a representation of a Sudoku puzzle as a sequence and needs to output the solution of the puzzle as a sequence. During the training, we provide information about a single cell using three tokens $(r, c, v(r, c))$: the first two tokens $(r, c)$ contain information about the position of the cell (row and column number) and the third token contains the number in that cell. Each training sequence is divided into two parts. The first part contains the information about cells whose values are given in the puzzle question and the second part contains information about unfilled cells in the solution. Note that there can multiple valid orders for the solution. Also, note that the length of the first part depends on the number of cells filled in the puzzle. We train the model using the next-token prediction loss but we don't apply the loss corresponding to the prediction of the filled cells given in the question.

We use a Transformer-based GPT-2 [RWC$^+$19] architecture with 8 layers for both puzzles. Each layer has 8 attention heads with a model dimension of 576 and an MLP of hidden dimension 3456 (6× model dimension) follows in each layer. The total number of parameters of our model is 42M. We use causal masking in the attention layers to avoid looking into the future.

**Evaluation metrics.** To evaluate the performance of our model, we use the following two metrics primarily: 1) **Cell accuracy**: denotes the percentage of the unfilled cells whose values are correctly predicted by the model. 2) **Complete puzzle accuracy**: denotes the percentages of the correctly solved puzzles in the evaluation dataset. A puzzle with even a single mistake is counted as incorrect.

## 3 Experiments on Sudoku puzzles

We study the performance of a Transformer model when the model is trained with the next-token prediction objective. We set up the model architecture and training of the model as discussed in Section 2 however, the question remains how to order cells in the input sequences of the Sudoku puzzle during the training of the model. Note that given the state of a Sudoku puzzle, some cells might be easier to solve than others so the order of the cells of input sequences provided during training could be important. We first try using a predefined fixed order or a random order of the cells during the training and inference in Section 3.1. However, this leads to poor performance. Thereafter, we turn our focus on using a solver to create a better order which we call solver-decomposed reasoning order (Section 3.2). Inspired by Chain-of-Thoughts literature [WWS$^+$22b], Section 3.3 uses solver-decomposed reasoning order only during the inference on the above trained models to provide position hints. Yet, conditioning on these position hints during decoding only provides a relatively small improvement in the performance showing that even if we tell the model to find the value in a particular cell, it has not learnt fully how to do so.

Therefore, in Section 3.4, we explore training the model using cells provided in solver-decomposed reasoning order. This provides a huge boost to the performance allowing the model to solve over 85% of the puzzles in the test set accurately. However, it still does not achieve near-perfect cell accuracy. Therefore, Section 3.5 uses beam search decoding to improve the performance.

### 3.1 Training using fixed or random order of the cells

A natural choice for the cell order in the input sequence would be to use a fixed order of the cells or a random order of the cells in the puzzle for the input sequence. Note that the order of the puzzle is only provided during the training and we do not penalize the model for wrong order during evaluation as long as it solves the given sudoku puzzle correctly.

**Fixed order of the cells.** In this ordering of the cells, we arrange the cells in a predefined fixed order of top-left to bottom-right of the board of the puzzle. To be more precise, for any two cells $(r, c)$ and $(r', c')$ where $r$ and $r'$ denote the row numbers and $c$ and $c'$ denote the column number, we will order the first cell $(r, c)$ before $(r', c')$ if $r < r'$ or $r = r'$ and $c < c'$. We order both parts of the puzzle (input sequence) - given cells in the puzzle and the remaining solution of the puzzle using the above-mentioned ordering.

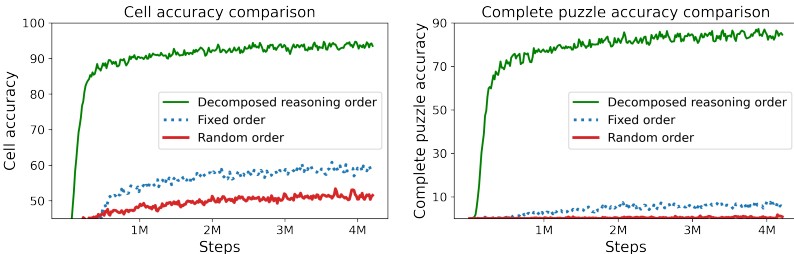

Figure 1: Comparison of cell accuracy and full puzzle accuracy for fixed order training, random order training and solver-decomposed reasoning order training.

**Random order of the cells.** Another way to arrange the cells that we consider is to randomly order cells in given cells of the puzzle and solution of the inputs. For any given prefix (state of the puzzle), we randomly pick a cell from the set of empty cells and append that cell and corresponding value to the prefix.

**Results.** We provide the experimental results for the fixed order in Figure 1. We see that the model trained with fixed order achieves 58.64% cell accuracy and only 7.2% full puzzle accuracy whereas the model trained with random order only achieves around 52% cell accuracy and only $1\%$ complete puzzle accuracy.

In the above ordering of the cells, given a state of the puzzle, the model decides on a random cell or fixed cell to output value but at that state, only a few cells might be easier to solve than others and the model trained using random or fixed order of cells do not necessarily decode the easier cells at that state. Therefore, inspired by Chain-of-thought literature [WWS+22b], we ask the following question: if we provide the model information during inference which cells are easier to fill then does the performance improve? Before we answer the above question, we define the solver-decomposed reasoning order which will be useful in finding cells that are easier to fill.

### 3.2 Solver-decomposed reasoning order

A natural way humans solve Sudoku is by iteratively trying to find cells that look easier to fill. The search process involves trying to see if any of a given set of strategies can be applied to fill in the value or otherwise make progress on a cell. Inspired by this analogy, we construct an order of filling cells using a solver. The solver uses 7 strategies as mentioned in Section 2. At any given state of the puzzle, it tries to apply to an easier strategy first and if it can not make progress with an easier strategy then it goes to a harder strategy. To apply a strategy, the solver goes through all the cells and tries to apply the strategy for each cell to make progress toward solving the puzzle. Progress doesn't necessarily mean filling a value in a cell but simply eliminating possible values from the candidate set (set of possible values) of a cell also counts as progress. We call the order given by the solver as *solver-decomposed reasoning order* or *decomposed reasoning order* for brevity when it is clear from the context. Note that the decomposed reasoning order arranges the cells based on how easy they are to fill in, as the solver initially employs simpler strategies to make progress.

### 3.3 Hinted cell accuracy

In recent years, chain-of-thought (CoT) prompting [WWS+22b] has emerged as one of the effective techniques to extract complex reasoning abilities from a model. The main idea of CoT is to lead the model to the correct output by providing intermediate steps to help the model. Inspired by the CoT prompting, we ask if providing the model additional information about easier-to-decode cells *during inference* improves the performance?

Specifically, we use decomposed reasoning order to provide position hints during inference. Recall that to infer a value at position $(r, c)$ at a state of the puzzle $s$, we provide $(s, r, c)$ as input to the transformer model and the random-order baseline model is trained to predict value $v$ as next token given positions in previous two-tokens $(r, c)$, and because positions are chosen randomly during the training, the model is forced to use the positions in previous two-tokens $(r, c)$ while predicting the

value. (Note that this is not the case for the model trained with fixed order and therefore, we don't consider them in this experiment).

Now, to provide additional information to the model about the easier cells to predict, we use the decomposed reasoning order. Specifically, for any given state $s$, we provide the state $s$ to the solver to obtain the position of the easiest cell. Suppose the solver picks $(r', c')$, then we provide $(s, r', c')$ to the trained model to predict a value at $(r', c')$. We reiterate this process for every non-empty cell of the puzzle. We measure the *cell accuracy* in this setting and we call this accuracy as **hinted cell accuracy** to denote the provided hints about easy-to-decode positions from the solver.

**Results.** We see that the model trained using random order achieves $54.57\%$ hinted cell accuracy. This means that providing hints about easy-to-decode positions improves the accuracy by around $3\%$ over without any hints. At first glance, it seems like the model is struggling significantly to implement the correct strategy even when we provide information about the positions of easy-to-fill cells as hints during the inference. However, this might not be the correct conclusion because of the following reasons: during the training, the model is trained to predict the value from random cells, and the model needs to learn and apply very hard strategies as well to improve its training loss. In the process of learning hard strategies, the model might fail to learn easier strategies as well because of various reasons (e.g., limited data corresponding to easier strategies, limited model size, etc.).

When we use decomposed reasoning order during the inference, it helps to improve the performance for the model trained using random-order of the cells but because the model needs to perform a hard search and reasoning task while decoding a value at a single cell, it not only seems to hurt the model in searching easy-to-decode cells but also affects its reasoning capabilities to decode a value at given cells even after we explicitly provide positions of easy to fill cells. This motivates us to use decomposed reasoning order during the training.

## 3.4 Using solver for CoT training

Solving the sudoku puzzle can be decomposed into two sub-tasks: 1) Search across the board to find the cells that are easy to fill and 2) After finding the easy-to-fill cell, apply the correct strategy on the cell to obtain a correct value in it. As mentioned earlier, the model trained for fixed-order and random-order does not have explicit incentives to perform a search for easy-to-fill cells. Therefore, the motivation for providing the solver-decomposed reasoning order during the training is to provide an order of cells such that training a model using the order helps the model to decompose the complex task of solving sudoku into smaller sub-tasks.

To provide decomposed reasoning order of cells during the training, we arrange the cells according to how easy to fill they are. Note that we can obtain this using Then, we use these sequences during the training with the next-token-prediction loss for all the tokens. Therefore, given a board state $s$, the loss corresponding to position tokens incentivizes the model to learn to find easy-to-decode cells, and the loss corresponding to value tokens incentivizes the model to learn the strategy.

**Result.** We provide the result for the decomposed reasoning order training in Figure 1. We see that using the decomposed reasoning order achieves the cell accuracy $94.23\%$ and complete puzzle accuracy $87.18\%$ accuracy. Training the model on the decomposed reasoning order improves cell accuracy by around $36\%$ over the fixed-order training and by around $43\%$ over the random-order training. The most noticeable improvement comes in complete puzzle accuracy where decomposed reasoning order training achieves $87.18\%$ accuracy whereas the fixed-order training achieves around $8\%$ accuracy and the random-order training achieves around $1\%$.

**Hinted accuracy for training using solver-decomposed reasoning order.** Even though solver-decomposed reasoning order training significantly improves performance over fixed-order training and random-order training, it does not achieve near-perfect accuracy. Therefore, to understand whether the model is struggling to perform a search for easy-to-decode training or to employ a strategy given a position, we perform the experiment of providing hints about easy-to-decode positions (presented in Section 3.3). Recall that to measure the hinted cell accuracy for a model, we provide information about easy-to-decode cells to the model during inference and measure cell accuracy in that setting. We see that the model with solver-decomposed reasoning order training achieves **99.02 %** hinted cell accuracy. This means that *the model can employ the correct strategy assuming it has access to information about easy-to-decode cells and that the performance gap in cell accuracy (94.23 % to near-perfect accuracy) is mainly due to searching for easy-to-decode cells.*

| | Cell accuracy | Complete puzzle accuracy |
|---|---|---|
| Beam width $k = 1$ | 94.23 % | 87.18 % |
| Beam width $k = 3$ | 96.07 % | 91.36 % |
| Beam width $k = 5$ | 98.03 % | 94.21 % |

Table 2: Performance (cell accuracy and complete puzzle accuracy) change as we increase beam-width in beam-search.

Next, we try to bridge the gap between $94.23\%$ cell accuracy and $99.02\%$ hinted cell accuracy achieved by the model trained using decomposed reasoning order. To improve the accuracy, we first understand the number of cells that are filled for a puzzle when the model is making the first mistake for the puzzle. Note that when the model makes a mistake by filling in an incorrect value on the puzzle, then the probability of the model making a mistake on the remaining empty cells increases. We also see this happen in our experiments (See Appendix E).

A hypothesis about lower cell accuracy than hinted cell accuracy is that given a certain state of the puzzle, the model might be confused between several cells about which cells are easier to decode. However, if the model is allowed to explore multiple potential cells of the puzzle, it might figure out the true solution as the model will make a prediction confidently for the true solution of the puzzle compared to other wrong solutions. Because of this reason, we try beam-search decoding for the model trained using solver-decomposed reasoning order.

### 3.5   Beam-search decoding

Beam-search decoding (used in many popular NLP systems, e.g. [WSC$^+$16]) in language modeling allows the model to explore multiple partial decoding of the sequences during the inference of a language model and output the most probable explored sequence. The beam width $k$ of the beam search decoding denotes how many partial sequences (hypotheses) are kept at each step. At each step of the decoding, it expands all partial solutions of the puzzle by decoding one more token. Then, among this expanded partial solution set, the model selects the top $k$ most probable partial solutions. This process is repeated for the decoding of every token. Note that beam search only maintains $k$ possible output sequences throughout the decoding process. Compared to standard decoding, beam search incurs a computational overhead of a factor of $k^2$. In the Sudoku puzzle, It is important to note that the beam search can not try out all possible outputs for the Sudoku puzzle. After all, the total number of outputs can be arbitrarily large because many of the empty cells will have on average 2 to 5 possible values and the total number of empty cells in the puzzle is at least 50.

**Results.** We present our results for beam-search decoding in Table 2. The beam search with $k = 1$ is equivalent to greedy decoding as it only keeps one partial sequence. We see that beam search with $k = 3$ improves the cell accuracy by around 2% and complete puzzle accuracy by around 4%. We see a similar improvement when we increase beam width from $k = 3$ to $k = 5$. Note that the cell accuracy with beam width $k = 5$ is able to bridge the gap from the hinted cell accuracy up to a large extent but does not need hints about easy-to-decode positions.

## 4   Analysis

In the above section, we showed that solver-decomposed reasoning order during the training can greatly help improve the model's performance. In this section, we analyze the trained model on several fronts. Section 4.1 contains a discussion about failure cases of the model in searching easy-to-decode cells. In Section 4.2, we show that the candidate set information emerges in the model to explain how the learned model is solving the puzzles. We compare the model's performance to a neural network-based method designed to solve the Sudoku puzzle [PPW18] in appendix F.1. Appendix F.2 contains the breakdown of the complete puzzle accuracy across various difficulties.

### 4.1   Failure in search for easy-to-decode cells

As discussed in section 3.4, the model trained using solver-decomposed reasoning order solves 94.23 % cells of the sudoku puzzles correctly. To understand the failure modes of the model, we measure the *hinted cell accuracy* by providing information about easy-to-decode cells to the model during inference. We see that the model achieves $99.02\%$ accuracy. This shows that the model can find

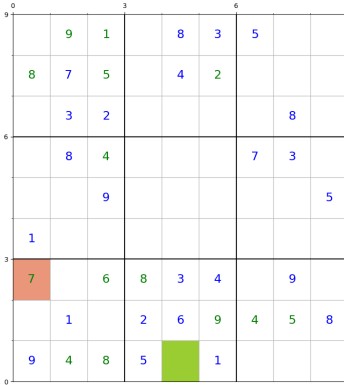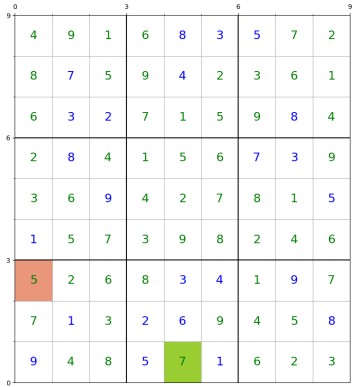

Figure 2: **A failure case of the model in searching for easy-to-decode cells.** The left figure shows the sudoku puzzle state when the model makes the first mistake and the right figure shows the puzzle's solution. Numbers given in the blue are provided in the puzzle. The puzzle makes a mistake by choosing to fill the red-colored cell whereas the green background cell can be easily filled.

the correct value at the cell when it is provided the information about easy-to-decode cells and the performance gap in cell accuracy is mainly due to the inability to search for easy-to-decode cells.

We also provide some examples of the puzzle situations in Figure 2 and Figure 7 (in Appendix) when the model makes a mistake by trying to fill a cell but there is another cell for which it is easier to fill. This supports our finding by hinted cell accuracy that the performance gap of our trained model to the perfect accuracy is due to the inability to search for easy-to-decode cells.

Additionally, a cell can be easy-to-decode because of either row, column or block constraint of the Sudoku puzzle. We found that our trained model misses more cells which are easy-to-decode because of the block constraint. This might be due to the input format being not explicit for block and explicit for row and column of a cell (recall that a cell is in the format of $(r, c, v(r, c))$ as input to the model). A natural extension in this case could be to provide a block number also as an input to the model. We leave it for future work.

## 4.2   Emergence of candidate set information in the model

We saw in the previous section that a model trained with puzzles given in solver-decomposed reasoning order performs very well. Therefore, we focus on how the model is learning such a task that requires planning and reasoning. As mentioned earlier, the sudoku solver (and to an extent humans) keeps track of possible values that a cell can take for the given puzzle. Therefore, we ask the following question: does the model also keep track of possible values of a cell? Can we extract them from the model?

We answer both of these questions (perhaps surprisingly) positively by showing that for a given puzzle, the candidate set of the solver can be extracted from the logits of the model. The candidate set of an empty cell keeps track of possible values that the cell can take given a state of the puzzle. Note that given some state of the puzzle, the candidate set at an empty cell $(r, c)$ can be different from $\{1, 2, \ldots, 9\} - \{$set of filled values in row $r$, column $c$ and box $b(r, c)\}$ as some of the strategy removes a value which does not occur in the same row, column or box.

**Calculating candidate set equivalence.** For all puzzles in the validation dataset, we obtain the candidate set of all empty cells from the solver when the number of filled cells is in the set $S = \{35, 40, 45, 50, 55, 60, 65, 70, 75\}$. For a state $s$ of a puzzle, we denote the candidate set of the solver at an empty cell $(r, c)$ as $f^*(s, r, c)$. We use $|f^*(s, r, c)|$ to denote the number of possible values in the candidate set $f^*(s, r, c)$. To extract the candidate set from the model at the state $s$ of the puzzle and at an empty cell $(r, c)$, we feed $(s, r, c)$ as the prefix to the model and values corresponding

| Number of filled cells | 35 | 40 | 45 | 50 | 55 | 60 | 65 | 70 | 75 |
|---|---|---|---|---|---|---|---|---|---|
| Accuracy (%) | 93.19 | 93.23 | 94.14 | 94.81 | 94.70 | 96.60 | 97.71 | 98.54 | 99.37 |

Table 3: Candidate set equivalence accuracy when the number of filled cells is different in the given puzzle. The candidate-set equivalence accuracy measures the average overlap between the solver's and the model's candidate set for the correctly solved puzzles.

to top-$k$ output logits where $k = |f^*(s, r, c)|$ becomes the candidate set of the model. We denote the model's candidate set as $m(s, r, c)$. Importantly, note that we DO NOT only evaluate the top-$k$ candidates on the cell the model chooses to predict. Although during its natural course of decoding the model might wish to decode cell location A, we force it (by conditioning) to decode at every other location and evaluate the top-$k$ candidates. This ensures that we are looking at what the model thinks is the set of possible candidates of cell location $(r_1, c_1)$ even when it has decided to decode cell $(r_2, c_2) \neq (r_1, c_1)$ next. We note that this style of probing differs from the more common way to perform a probing analysis which involves learning a linear/non-linear probe which takes in the embedding and outputs a label indicating a concept. However, we use probing in more general sense to refer to understand some of the inner workings of the model.

The accuracy for the candidate set equivalence between the solver and the model at a state $s$ of a puzzle and at an empty $(r, c)$ is measured by $|f^*(s, r, c) \cap m(s, r, c)|/|f^*(s, r, c)|$. The reported accuracy at position $n \in S$ in Table 3 is the average over all empty cells when the number of filled cells is $n$ for the puzzles which are correctly solved by the model. Intuitively, the candidate-set equivalence accuracy measures the average overlap between solver's and model's candidate set for the correctly solved puzzles.

**Results.** The results of candidate set equivalence accuracy are given in Table 3. We see that for all positions the average overlap between the solver's and the model's candidate set is above 93 %. This overlap improves to around 96.5 % when the prefix has information about 60 cells and to around 98.5 % when the prefix contains information about 70 cells. Note that to extract the candidate set of the model, we are just reading the logits and not even training a linear function. Additionally, the candidate set equivalence result is not only for cells that are easy to decode but for all empty cells. Moreover, during the training of the model, no direct information about the candidate set is provided and the model is only trained to predict the correct value for a cell and therefore is not directly incentivized to predict the correct candidate set for *all* the empty cells with such a high accuracy.

## 5 Conclusion

We have shown that even on complex logical reasoning tasks such as Sudoku and Zebra puzzles, simple next-token prediction provided with a high-level decomposition of the reasoning steps during training is able to learn to solve the task. This suggests that, given the right level of detail and breakdown of reasoning steps in the training data, a pre-trained model might already present as a strong reasoning engine (without the need for post-training techniques such as fine-tuning, prompt engineering, self-consistency, tree-of-thoughts etc). These techniques might help significantly boost the baseline performance of a model or potentially make up for deficiencies in the pre-training data however. To move towards more general reasoning systems, an interesting challenge to overcome would be to simulate the decomposed reasoning data in an efficient manner. These tasks capture many different types of constraint satisfaction problems and we believe the framework and results should generalize to other settings as well.

Finally, we conclude with some limitations of our study. Firstly, we note that we studied a synthetic setting on a toy task and real-world reasoning and planning tasks can be much more abstract and challenging. More specifically, Sudoku is a task which doesn't require the same degree of long-term planning as some harder benchmarks. That is, any cell we can make progress on is progress unlike constraint problems where one might need to backtrack. Moreover, we focused on a reasoning setting where creative thinking was not required. That is, the model did not need to invent new strategies to solve any test time puzzle. It is an interesting future direction to study to what extent causal language modeling can yield novel reasoning strategies. Moreover, there can be many different types of reasoning tasks which are not logic puzzles (for instance probabilistic puzzles or rule-less puzzles, see e.g. [GLFS24]) and our experiments do not explore those.

## Acknowledgments and Disclosure of Funding

Authors would like to thank Erik Vee for guiding them to use the hinted cell accuracy to understand the failure modes of the trained model.

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

# A  Example of a Zebra puzzle

*There are 3 people next to each other in a row. Everyone has a different name: Ali, Rose, Randy. Every one lives in a different colored house: gold, silver, indigo. Everyone likes a different drink: orange juice, beer, coffee. Match the people to the correct value for each of their characteristics using the clues.*

1. *The person who likes orange juice is immediately to the left of the person who likes coffee.*
2. *The person who likes beer is somewhere to the left of the person who lives in the indigo house.*
3. *The person at the 1st position is Rose.*
4. *Randy is not the person who likes orange juice.*
5. *Randy is the person who lives in the gold house.*

Figure 3: An example Zebra puzzle with 3 entities, each having 3 attributes.

# B  Related work

There are many works which study the ability of language models to perform reasoning tasks which involve search and planning with mixed evidence as to whether they are actually learning to reason and plan. [BMR+20] was a seminal work which showed that large language models (LLMs) are few-shot learners and [KGR+22] argued that they can be zero-shot reasoners. [LAD+22] shows that by fine-tuning on the appropriate data LLMs can exhibit a high performance on the non-trivial MATH [HBK+21] dataset. In addition, the reports of frontier models like GPT-4 [AAA+23] and Gemini [TAB+23] also contain support for the idea that LLMs can perform reasoning and planning. Building on these lines of work, [HXX+22, SBM+23, ABB+22, SWW+23] employ LLMs in planning tasks in the robotics domain.

A number of follow-up works study how we can improve the reasoning and planning capabilities of LLMs using various techniques such as prompt engineering, tool use, using LLMs in combination with an external deduction engine. Some of the prominent works in this bracket are Chain-of-Thought prompting [WWS+22b], least-to-most prompting [ZSH+22], self-consistency [WWS+22a], tree-of-thought prompting [Lon23, YYZ+24, XKZ+24], program of thoughts [CMWC22], planning for code generation [ZCS+23]. Of these, [Lon23] evaluate the efficact of Tree-of-Thought reasoning using LLMs like GPT-4 on solving Sudoku puzzles and only achieve results on 5x5 Sudoku puzzles, Moreover, the tree-of-thought prompting technique is known to quite expensive to run. [ZLH24] use LLMs to provide a commonsense world model and a policy which can be fed to a Monte-Carlo tree search algorithm. The ReAct framework [YZY+22] uses LLMs to generate reasoning traces and task-specific actions in an interleaved manner.

In contrast to the above, [VOSK22] show that LLMs when acting alone or when combined with techniques such as Chain-of-Thought or Tree-of-thought cannot solve some standard planning and reasoning benchmarks when the questions are rephrased with a new terminology. This is even when we use techniques such as Chain-of-Thought, fine-tuning etc. [XZC+24] show that even the biggest LLMs perform very poorly at real-world travel planning tasks with a multitude of soft and hard constraints. [DLS+24] show that LLMs are limited and brittle in their ability to perform compositional tasks such as multi-digit multiplication, logic grid puzzles and dynamic programming. [MHF+23] argue that LLMs have weak cognitive maps which are crucial for planning. [BN24] show that rather than the architecture, the training objective of next-token prediction might be crippling the planning and reasoning ability of a language model.

There are many works which use the help of synthetic tasks to gain insights into how Transformer language models work. We present a non-exhaustive list here. [LHB+23] use the synthetic task of learning to predict valid next moves in an Othello game to study whether an internal model of the Othello board emerges in the model or not. [AZL23] use synthetic tasks such as learning context-free grammars to understand the mechanics of how large-scale learning in LLMs works. [GTLV22] use

linear regression from in-context examples to study the in-context learning phenomenon. [NCL$^+$23] use the task of modular addition to understand the specific algorithm a shallow Transformer implements to solve the problem. [AG23] use the task of sorting a list of numbers to study length generalization.

**Comparison to traditional solvers and other ML approaches.** Traditional constraint satisfaction libraries use very powerful combinatorial search algorithms to solve logic puzzles and are much more powerful than any deep model we learn here. In addition, many prior works study machine learning-based approaches for solving general combinatorial problems [BPL$^+$16, MSIB21, CFK$^+$23]. In addition, there are several approaches that tend to handcraft the architecture or loss to the puzzle using human understanding of the puzzle structure [MKPZ11, PPW18, Zhu]. Even though our goal is to understand the capabilities and limitations of causal language modeling and not to compete with such solvers, we discuss some of these works more in detail.

[MKPZ11] try to setup a Hopfield network to solve Sudoku puzzles. [PPW18] handcrafts the recurrent network to match the puzzle structure (and obey the constraints) and performs multiple rounds of message passing between cells of the sudoku puzzle to arrive at a solution. We evaluate our trained model (trained using causal language modeling) on the test dataset proposed in this work and we observe a comparable performance without handcrafting the network or loss function. [Zhu] achieves a 65% accuracy of RNN based solvers on 3x3 Sudoku puzzles. [NB21] study how well GPT-2 models trained on natural language perform on puzzle tasks such as Rubik's cube and Sudoku. [YIL23] study solving Sudokus using a recurrent form of Transformers by baking the knowledge of Sudoku's constraints into the model architecture and training pipeline. For chess, works like [RDM$^+$24] use a chess engine such as Stockfish to provide supervised labels for different board states and train a Transformer network to predict the value function of a board state. This can then be used to play expert level chess.

## C   Details about our list of strategies

As mentioned earlier, we consider puzzles with 7 strategies for both Sudoku and Zebra puzzles.

### C.1   Strategies for Sudoku puzzles

We first list all the strategies used in Sudoku puzzles with an explanation.

1. **Lone single**: This strategy is applied to a cell where only one candidate number is possible based on the rules.
2. **Hidden single**: This strategy is applied the situation where a number can only be placed in one specfic cell within a row, column or box.
3. **Naked pair**: This strategy is applied when two cells in a row, column or box contain the exact same two admissible numbers. This strategy is used to eliminate the number of possible values.
4. **Naked triplet**": This strategy is applied when three cells in a row, column or box contain the exact same three admissible numbers. Similar to the "naked pair" strategy, this strategy is used to eliminate the number of possible values.
5. **Locked candidate**: This strategy is applied when all the possible positions for a specific number within a box are on the same row or column.
6. **XY wing**: This is a complicated strategy that involves three cells and multiple deduction steps. First, identify a vacant cell (called a pivot) that has two admissible numbers (denoted by $X$ and $Y$); second, identify two other cells (called wing cells) such that each of them shares a column, row or box with the pivot, and one cell has two admissible numbers $X$ and $Z$, and the other cell has two admissible numbers $Y$ and $Z$. third, for every other cell that share a column, row or box with both wing cells, $Z$ can be eliminated from their admissible numbers.
7. **Unique rectangle**: This is another complicated strategy that involves four cells. First, identify four cells that forms a rectangle such that three of these cells have only two

admissible numbers and the numbers are the same, and the fourth cell share at least of the numbers as an admissible number; second, both numbers can be eliminated from the admissible numbers for the fourth cell.

We provide some visual examples of complex strategies in Figure 4.

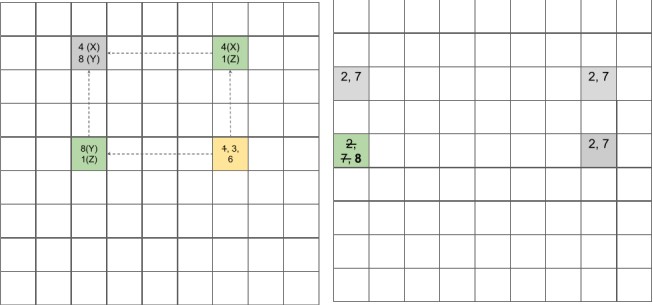

Figure 4: Examples of complex strategies that involves reasoning about multiple cells. Left: XY-Wing, where a pivot cell (gray) has two candidate values (X and Y), the wing cells (green) share a column, row or box with the pivot and share one candidate value (X or Y) with pivot and another common candidate value (Z), then in any cell that shares a column, row or box with both wing cells (yellow), we can eliminate Z from the candidate set; Right: Unique Rectangle, where four cells form a rectangle, among which three cells (gray) share the exact same 2 candidate values, and the fourth cell (green) share at least one of the 2 values, then both values can be eliminated from the candidate set for the fourth cell.

## C.2 Relationtypes for Zebra puzzles

We now list all the relationship types for the Zebra puzzles. The examples of the following relation types are given for the original Zebra puzzles.

1. **Is equal to**: This relation type provides the value for an attribute. An example of this type of clue can be the Norwegian lives in the first house.

2. **Is not equal to**: This relation type provides the information that an attribute can not have a particular value. An example of this type of clue can be the Englishman does not live in the first house.

3. **Immediate left**: This relation type provides the relation order for the values either between attribute values or entities in the solution. An example of this type of clue can be the person with the dog is immediately left of the person who drinks the coffee.

4. **Neighbour of**: This relation type provides the information that an entity with a particular attribute value is the neighbor of another entity. This relation type generalizes the "immediate left" relationship to include the immediate neighbors of the left and right sides. An example this type of clue is the person with a dog is next to the person who drinks milk.

5. **Ends in**: This relation type provides the information that an entity with the particular attribute value is on either end of the order. For example, the person with the Zebra is on either end of the order.

6. **Left of**: This relation type provides the relative order of two entities with some particular values. Note that left-of relation does not mean the immediate left of an entity. For example, the person who drinks tea is left of the Japanese person.

7. **In between**: This relation type provides the relative order of three entities with some particular attribute values. For example, the Englishman lives in-between the person with the Horse and the person who drinks coffee.

# D   Extended preliminaries

In this section, we provide additional details about the setup of the Zebra puzzle, dataset, and hyperparameters.

**Zebra puzzle and solver.** The Zebra puzzle is characterized by the number of entities $m$ and the number of attributes $n$ for each entity. e.g., in Figure 3 each person is an entity, and name, house color and drink are attributes associated with each entity. The relationships between entities and attribute values are given as clues in the puzzle and the task is to figure out values for all attributes and all the entities. See Figure 3. Observe that each clue puts some constraints on the value of attributes and entities. e.g., "The person who likes beer is somewhere to the left of the person who lives in the indigo house." clue says that the house (entity) which has drink attribute = beer is somewhere left to the house whose color attribute = indigo. As we allow more and more complex relationships in clues, the puzzles become more and more complex. In addition, increasing size makes the puzzles more complex as well as more and more interconnected clues are required to uniquely pin down a solution. Larger puzzles also have a higher chance of deeper and trickier reasoning chains being utilized. Similar to Sudokus, a generalized version of $m \times n$ Zebra puzzles is also NP-hard. Unlike Sudoku puzzles, where the constraints are only the uniqueness constraints within each row, column and box, Zebra puzzles can have a much more diverse set of constraints which significantly increases the number of 'strategies' that can be used to make progress. Moreover, Zebra puzzles are a step closer to natural language than Sudoku puzzles.

In the Zebra puzzles, we have 7 different types of clues. Details about each of the clue types is provided in Appendix C. We generate our own dataset of Zebra puzzles as follows. We first create a Zebra puzzle solver. The solver for the Zebra puzzle takes in a clue set and iteratively tries to make progress by using $k$-sized subsets of the clues at a time (for $k$ ranging from 1 to 3). If it is able to make a deduction, the solver marks that entry in the answer table and iterates over the clue subsets again. Given this solver, a new puzzle is generated by starting with an empty clue set and iteratively adding randomly generated clues until the solver is able to successfully solve the puzzle. While adding new clues we ensure we do not add duplicates. Nonetheless, some clues might still be rendered redundant due to the presence of 2 or more other clues. To keep the clue set lean, once we have a puzzle that the solver is able to solve, we filter out the clues unused by the solver. We generate puzzles of sizes $m \times n$ for $m, n$ ranging in $[3, 4, 5, 6]$.

Our training dataset for the Zebra experiment contains 0.3M puzzles and the test dataset contains 15k puzzles. The input to the model during the training is divided into two parts. The first part (given in the puzzle) contains the clues and the second part (solution) contains values for all attributes and all entities. Each clue contains two parts: 1) the relationship type between attributes and entities and 2) the specific attributes and entity values that are in this relationship whereas the solution part of the puzzle consists of multiple triplets of entity, attribute, and the solution for that entity and attribute. Similar to the sudoku puzzle, there can be different orders in which the solution triplets for each entity and attribute can be provided during the training.

**Dataset.** We consider the Sudoku dataset from [Rad20] and then we adapt a Sudoku solver from [MP23] to filter out the puzzles that can not be solved by the 7 strategies listed above. After filtering, the dataset contains 1.9M puzzles. We randomly choose 0.1M puzzles from these puzzles and use them as a validation dataset for the evaluation of the model and the remaining 1.8M puzzles are part of our training dataset.

We use the AdamW optimizer for our experiments. For all the experiments, learning rate is set to $1 \times 10^{-4}$ and models are trained for 4 million steps with a batch size of 64. We use the cosine learning rate schedule [LH16] with the first 4000 tokens as the warmup phase and an end learning rate factor of 0.2.

# E    Mistake position frequency experiment.

We present the results about the mistake frequency and first mistake frequency in Figure 5. In this section, we show that for a puzzle, the model makes more first mistakes for that puzzle at the start of the puzzle when there are more empty cells because for a model, it is harder to predict the correct value for that cell but the distribution of all mistakes is more towards the later mistakes. This shows that when a model makes a mistake on a sequence, it is likely that it will keep making a mistake because of the invalid prefix.

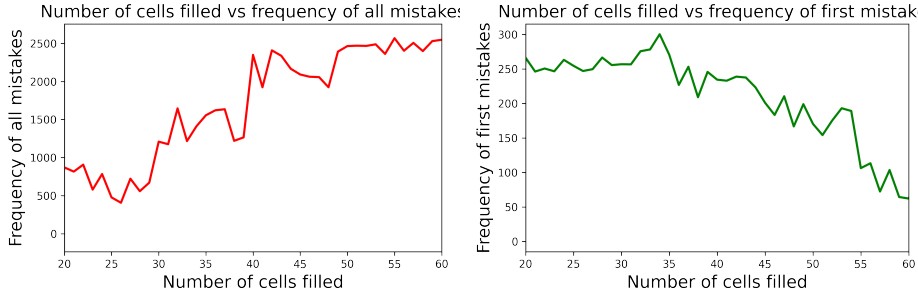

Figure 5: Left figure: Plots the number of mistakes made after how many number of cells were filled. Right figures: Plots the number of first mistakes that are made against number of cells that were filled when it made the first mistake.

## F    Performance analysis of the model

### F.1    Comparison with neural network-based Sudoku solver

The goal of our work is to understand the capabilities and limitations of causal language modeling (and not to propose a new approach to solve the Sudoku puzzle). To understand if there exists a performance gap between a model trained using causal language modelling and a model specifically designed to solve Sudoku puzzles, we compare the performance of our model with a neural network based Sudoku solver proposed in [PPW18]. Palm et al. [PPW18] handcrafts the recurrent network to match the Sudoku puzzle structure (and obey the constraints) and perform multiple rounds of message passing between cells of the Sudoku puzzle to arrive at a solution. We evaluate our trained model on the test dataset proposed in [PPW18] of 18000 Sudoku puzzles (See Table 4 for the result). We observe that our trained model (trained using solver-decomposed reasoning order and causal language modeling) combined with the beam search obtains a comparable performance *without handcrafting the network or loss function*.

|                            | Eval accuracy | Eval complete puzzle accuracy |
| -------------------------- | ------------- | ----------------------------- |
| Beam search width=1        | 98.15 %       | 94.76 %                       |
| Beam search width=3        | 98.28 %       | 95.12 %                       |
| Beam search width=5        | 98.37 %       | 95.43 %                       |
| Recurrent Relational Network | NA          | 96.6 %                        |

Table 4: Evaluating our model on the evaluation dataset of Sudoku given in Recurrent Relational Network (RRN) by Palm et al. [PPW18]. Our trained model performs comparably to the RRN model but *does not require handcrafting the network and training procedure* for training on the Sudoku puzzles.

### F.2    Performance analysis of the model using the difficulty of the puzzles

We provide the breakdown of the complete puzzle accuracy across various difficulties in Figure 6 to better understand the performance of the trained model. We use the difficulty measure provided in the Kaggle dataset [Rad20]. To obtain the difficulty of a puzzle, it considers a solver (different from the one we use to generate our solver-order data) which tries to iteratively make progress on a puzzle using some simple strategies. When the solver gets stuck, it makes guesses and tries to solve the puzzle. The difficulty rating is the average number of guesses the solver had to make to solve the puzzle. We wish to point out that even a puzzle rated 1.0 can require complex strategies beyond simple scanning to solve them without guessing and therefore, this is an imperfect measure of the difficulty.

In Figure 6, we observe that the model achieves almost perfect complete puzzle accuracy for lower difficulty accuracy and as the difficulty of the puzzle increases, the complete puzzle accuracy goes down. We want to note that even when the difficulty rating is between 3 to 3.5, the model can solve around 50 % of the puzzles completely. Additionally, the advantage of beam search increases for

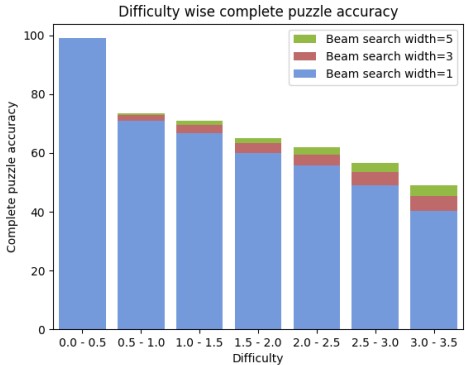

Figure 6: **Complete puzzle accuracy for different difficulty Sudoku puzzles.** The difficulty rating is computed as the average number of guesses the rating-solver had to make to solve the puzzle therefore, the difficulty rating is an imperfect measure of the difficulty.

the higher difficulty puzzles as the model can explore multiple solutions when it has confusion and output a solution at the end.

# G   Additional examples of failure mode of the model

# H   An example of candidate set for the puzzle

### H.1   Sudoku puzzle

A generalized version of Sudoku has a board of size $n \times n$ with $n$ boxes of size $\sqrt{n} \times \sqrt{n}$, and the goal of the game is to fill a partially filled board so that each row, column and $\sqrt{n} \times \sqrt{n}$ boxes contains a full set of numbers from $1$ to $n$. Implicitly, this means that the goal is to fill out the complete board such that none of the rows, columns, and boxes contain duplicates. In our experiments, we consider Sudoku of board size $9 \times 9$ which is further divided into nine $3 \times 3$ boxes.

### H.2   Candidate set example

The candidate set for a position $(r, c)$ keeps track of all possible values that the cell can take (See Section 4.2 for more details) and then uses the candidate sets to either deduce a value at a particular cell or narrow down the candidate set (set of possible values) at an empty cell.

# I   Experiments on Zebra puzzle

To extend our results beyond Sudoku, we also conduct our experiments on the Zebra puzzle (also known as Einstein's Puzzle). Like Sudoku puzzles, we consider providing the solution in either a fixed, random, or solver-decomposed reasoning order during the training.

**Order of the solution**    The input provided for a Zebra puzzle during training consists of two parts: the clues and the solution. The solution part contains values assigned to each entity and attribute. Based on the given clues, certain values for specific entities and attributes are easier to determine than others (e.g., in Figure 3, the third clue immediately reveals that the first house has a person with the name attribute = Rose). Thus, as seen in the Sudoku puzzle, the order in which the solution is provided is important.

Similar to Sudoku puzzles, we consider providing the solution in either a fixed, random, or solver-decomposed reasoning order during the training. In all cases, the clues part of the input remains unchanged. In fixed-order training, the solution is given in a predetermined sequence (we use the order starting from the first house's first attribute to the last house's first attribute, followed by the

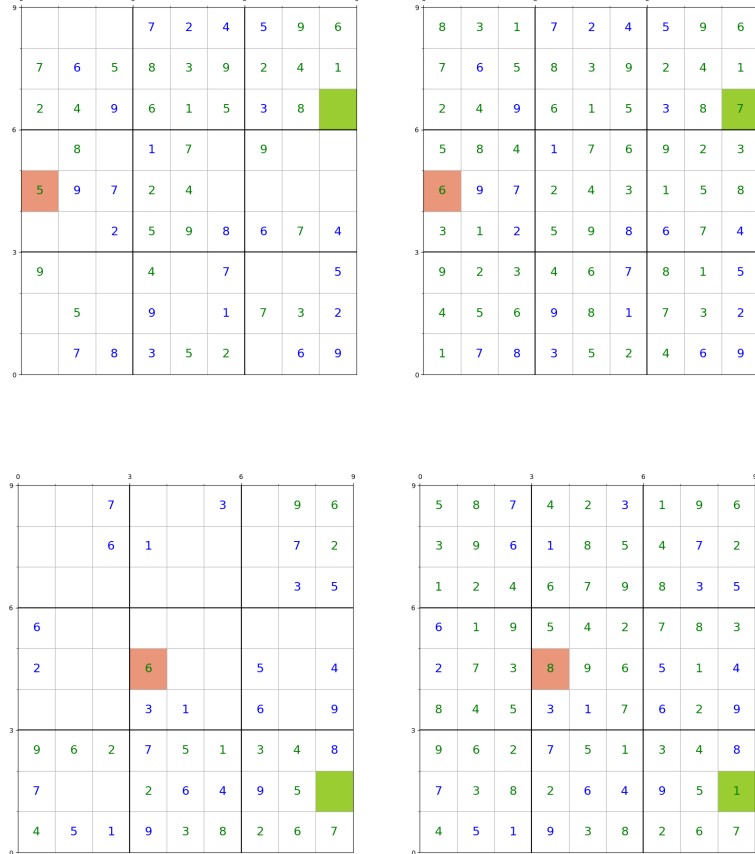

Figure 7: **Additional examples of the failure of the model in searching for easy-to-decode cells.** Both left figures show the sudoku puzzle state when the model makes the first mistake and both right figure shows the corresponding puzzle's solution. Numbers given in the blue are provided in the puzzle. The puzzle makes a mistake by choosing to fill the red-colored cell whereas the green background cell can be easily filled.

next attribute). In random-order training, the solution is shuffled. In solver-decomposed reasoning order, the solution is arranged based on how the solver approaches the puzzle, progressively using smaller subsets of clues to make progress and therefore, dividing the reasoning to solve the puzzle into multiple stages.

**Results**  We plot the cell accuracy and complete puzzle accuracy on an evaluation set of 1k puzzles during the training in Figure 9. We see that the training using solver-decomposed reasoning order achieves 95.63 % cell accuracy and 91.17 % complete puzzle accuracy whereas the random order training achieves almost zero complete puzzle accuracy and the fixed order training achieves 79.36 % complete puzzle accuracy. We believe this is due to a larger number of small-sized Zebra puzzles in the evaluation set (e.g., puzzles with 3 or 4 attributes and entities) which are easier to solve than larger-sized Zebra puzzles. We also evaluate the model's performance by using beam search decoding and report the results in Table 5. We see that using beam search decoding with width=3 improves the performance by 0.7 % and increasing it to width=5 improves the performance by an additional 0.2 %.

Figure 8: An example of the candidate set for a puzzle

|  | Evaluation accuracy | Eval complete puzzle accuracy |
|---|---|---|
| Beam search width=1 | 95.63 % | 91.17 % |
| Beam search width=3 | 96.03 % | 91.83 % |
| Beam search width=5 | 96.26 % | 92.04 % |

Table 5: **Zebra puzzle results.** The training dataset contains Zebra puzzles with the no. of entities and the no. of attributes in $\{3, 4, 5, 6\}$ set. For each combination of the no. of entries and attributes, we generate 20k puzzles therefore, the complete dataset contains 320k puzzles of varying sizes. From the complete dataset, we randomly choose 15k puzzles for evaluation and the rest of the puzzles for training the model. Evaluation accuracy: the percentage of correctly predicted attributes on the evaluation set and eval complete puzzle accuracy: the percentage of correctly and completely solved puzzles.


Figure 9: Comparison of cell accuracy and full puzzle accuracy for fixed order training, random order training, and solver-decomposed order training

- The abstract and/or introduction should clearly state the claims made, including the contributions made in the paper and important assumptions and limitations. A No or NA answer to this question will not be perceived well by the reviewers.
- The claims made should match theoretical and experimental results, and reflect how much the results can be expected to generalize to other settings.
- It is fine to include aspirational goals as motivation as long as it is clear that these goals are not attained by the paper.

2. **Limitations**

Question: Does the paper discuss the limitations of the work performed by the authors?

Answer: [Yes]

Justification: The limitations are discussed in Section 5.

Guidelines:

- The answer NA means that the paper has no limitation while the answer No means that the paper has limitations, but those are not discussed in the paper.
- The authors are encouraged to create a separate "Limitations" section in their paper.
- The paper should point out any strong assumptions and how robust the results are to violations of these assumptions (e.g., independence assumptions, noiseless settings, model well-specification, asymptotic approximations only holding locally). The authors should reflect on how these assumptions might be violated in practice and what the implications would be.
- The authors should reflect on the scope of the claims made, e.g., if the approach was only tested on a few datasets or with a few runs. In general, empirical results often depend on implicit assumptions, which should be articulated.
- The authors should reflect on the factors that influence the performance of the approach. For example, a facial recognition algorithm may perform poorly when image resolution is low or images are taken in low lighting. Or a speech-to-text system might not be used reliably to provide closed captions for online lectures because it fails to handle technical jargon.
- The authors should discuss the computational efficiency of the proposed algorithms and how they scale with dataset size.
- If applicable, the authors should discuss possible limitations of their approach to address problems of privacy and fairness.
- While the authors might fear that complete honesty about limitations might be used by reviewers as grounds for rejection, a worse outcome might be that reviewers discover limitations that aren't acknowledged in the paper. The authors should use their best judgment and recognize that individual actions in favor of transparency play an important role in developing norms that preserve the integrity of the community. Reviewers will be specifically instructed to not penalize honesty concerning limitations.

3. **Theory Assumptions and Proofs**

Question: For each theoretical result, does the paper provide the full set of assumptions and a complete (and correct) proof?

Answer: [NA]

Justification: No theoretical results.

Guidelines:

- The answer NA means that the paper does not include theoretical results.
- All the theorems, formulas, and proofs in the paper should be numbered and cross-referenced.
- All assumptions should be clearly stated or referenced in the statement of any theorems.
- The proofs can either appear in the main paper or the supplemental material, but if they appear in the supplemental material, the authors are encouraged to provide a short proof sketch to provide intuition.
- Inversely, any informal proof provided in the core of the paper should be complemented by formal proofs provided in appendix or supplemental material.
- Theorems and Lemmas that the proof relies upon should be properly referenced.

4. **Experimental Result Reproducibility**

Question: Does the paper fully disclose all the information needed to reproduce the main experimental results of the paper to the extent that it affects the main claims and/or conclusions of the paper (regardless of whether the code and data are provided or not)?

Answer: [Yes]

Justification: All important details of the experiment settings are discussed and presented.

Guidelines:

- The answer NA means that the paper does not include experiments.
- If the paper includes experiments, a No answer to this question will not be perceived well by the reviewers: Making the paper reproducible is important, regardless of whether the code and data are provided or not.
- If the contribution is a dataset and/or model, the authors should describe the steps taken to make their results reproducible or verifiable.
- Depending on the contribution, reproducibility can be accomplished in various ways. For example, if the contribution is a novel architecture, describing the architecture fully might suffice, or if the contribution is a specific model and empirical evaluation, it may be necessary to either make it possible for others to replicate the model with the same dataset, or provide access to the model. In general. releasing code and data is often one good way to accomplish this, but reproducibility can also be provided via detailed instructions for how to replicate the results, access to a hosted model (e.g., in the case of a large language model), releasing of a model checkpoint, or other means that are appropriate to the research performed.
- While NeurIPS does not require releasing code, the conference does require all submissions to provide some reasonable avenue for reproducibility, which may depend on the nature of the contribution. For example
  - (a) If the contribution is primarily a new algorithm, the paper should make it clear how to reproduce that algorithm.
  - (b) If the contribution is primarily a new model architecture, the paper should describe the architecture clearly and fully.
  - (c) If the contribution is a new model (e.g., a large language model), then there should either be a way to access this model for reproducing the results or a way to reproduce the model (e.g., with an open-source dataset or instructions for how to construct the dataset).
  - (d) We recognize that reproducibility may be tricky in some cases, in which case authors are welcome to describe the particular way they provide for reproducibility. In the case of closed-source models, it may be that access to the model is limited in some way (e.g., to registered users), but it should be possible for other researchers to have some path to reproducing or verifying the results.

5. **Open access to data and code**

   Question: Does the paper provide open access to the data and code, with sufficient instructions to faithfully reproduce the main experimental results, as described in supplemental material?

   Answer: [Yes]

   Justification: The GitHub link of the code is provided.

   Guidelines:

   - The answer NA means that paper does not include experiments requiring code.
   - Please see the NeurIPS code and data submission guidelines (`https://nips.cc/public/guides/CodeSubmissionPolicy`) for more details.
   - While we encourage the release of code and data, we understand that this might not be possible, so "No" is an acceptable answer. Papers cannot be rejected simply for not including code, unless this is central to the contribution (e.g., for a new open-source benchmark).
   - The instructions should contain the exact command and environment needed to run to reproduce the results. See the NeurIPS code and data submission guidelines (`https://nips.cc/public/guides/CodeSubmissionPolicy`) for more details.
   - The authors should provide instructions on data access and preparation, including how to access the raw data, preprocessed data, intermediate data, and generated data, etc.
   - The authors should provide scripts to reproduce all experimental results for the new proposed method and baselines. If only a subset of experiments are reproducible, they should state which ones are omitted from the script and why.
   - At submission time, to preserve anonymity, the authors should release anonymized versions (if applicable).
   - Providing as much information as possible in supplemental material (appended to the paper) is recommended, but including URLs to data and code is permitted.

6. **Experimental Setting/Details**

   Question: Does the paper specify all the training and test details (e.g., data splits, hyper-parameters, how they were chosen, type of optimizer, etc.) necessary to understand the results?

   Answer: [Yes]

   Justification: Details provided in the experiments section.

   Guidelines:

   - The answer NA means that the paper does not include experiments.
   - The experimental setting should be presented in the core of the paper to a level of detail that is necessary to appreciate the results and make sense of them.
   - The full details can be provided either with the code, in appendix, or as supplemental material.

7. **Experiment Statistical Significance**

   Question: Does the paper report error bars suitably and correctly defined or other appropriate information about the statistical significance of the experiments?

   Answer: [No]

   Justification: Factors which can cause variability in our results include random seed used for optimization, train/test split. Due to limited computational and temporal resources we do not report error bars from multiple runs with different random seeds. However, we observed during our iterations to try and find the best hyper-parameter settings for our experiments, all runs were stable and results were robust and consistent across runs with identical hyper-parameters.

   Guidelines:

   - The answer NA means that the paper does not include experiments.

- The authors should answer "Yes" if the results are accompanied by error bars, confidence intervals, or statistical significance tests, at least for the experiments that support the main claims of the paper.
- The factors of variability that the error bars are capturing should be clearly stated (for example, train/test split, initialization, random drawing of some parameter, or overall run with given experimental conditions).
- The method for calculating the error bars should be explained (closed form formula, call to a library function, bootstrap, etc.)
- The assumptions made should be given (e.g., Normally distributed errors).
- It should be clear whether the error bar is the standard deviation or the standard error of the mean.
- It is OK to report 1-sigma error bars, but one should state it. The authors should preferably report a 2-sigma error bar than state that they have a 96% CI, if the hypothesis of Normality of errors is not verified.
- For asymmetric distributions, the authors should be careful not to show in tables or figures symmetric error bars that would yield results that are out of range (e.g. negative error rates).
- If error bars are reported in tables or plots, The authors should explain in the text how they were calculated and reference the corresponding figures or tables in the text.

8. **Experiments Compute Resources**

Question: For each experiment, does the paper provide sufficient information on the computer resources (type of compute workers, memory, time of execution) needed to reproduce the experiments?

Answer: [Yes]

Justification: Details are provided in the experiments section of the paper.

Guidelines:
- The answer NA means that the paper does not include experiments.
- The paper should indicate the type of compute workers CPU or GPU, internal cluster, or cloud provider, including relevant memory and storage.
- The paper should provide the amount of compute required for each of the individual experimental runs as well as estimate the total compute.
- The paper should disclose whether the full research project required more compute than the experiments reported in the paper (e.g., preliminary or failed experiments that didn't make it into the paper).

9. **Code Of Ethics**

Question: Does the research conducted in the paper conform, in every respect, with the NeurIPS Code of Ethics https://neurips.cc/public/EthicsGuidelines?

Answer: [Yes]

Justification: We do not conduct research using human subjects. Our work is foundational and does not directly pose a risk of negative societal impact. It helps develop our understanding of today's Machine Learning models.

Guidelines:
- The answer NA means that the authors have not reviewed the NeurIPS Code of Ethics.
- If the authors answer No, they should explain the special circumstances that require a deviation from the Code of Ethics.
- The authors should make sure to preserve anonymity (e.g., if there is a special consideration due to laws or regulations in their jurisdiction).

10. **Broader Impacts**

Question: Does the paper discuss both potential positive societal impacts and negative societal impacts of the work performed?

Answer: [NA]

Justification: Paper is foundational research with no tie with any real-world application.

Guidelines:

- The answer NA means that there is no societal impact of the work performed.
- If the authors answer NA or No, they should explain why their work has no societal impact or why the paper does not address societal impact.
- Examples of negative societal impacts include potential malicious or unintended uses (e.g., disinformation, generating fake profiles, surveillance), fairness considerations (e.g., deployment of technologies that could make decisions that unfairly impact specific groups), privacy considerations, and security considerations.
- The conference expects that many papers will be foundational research and not tied to particular applications, let alone deployments. However, if there is a direct path to any negative applications, the authors should point it out. For example, it is legitimate to point out that an improvement in the quality of generative models could be used to generate deepfakes for disinformation. On the other hand, it is not needed to point out that a generic algorithm for optimizing neural networks could enable people to train models that generate Deepfakes faster.
- The authors should consider possible harms that could arise when the technology is being used as intended and functioning correctly, harms that could arise when the technology is being used as intended but gives incorrect results, and harms following from (intentional or unintentional) misuse of the technology.
- If there are negative societal impacts, the authors could also discuss possible mitigation strategies (e.g., gated release of models, providing defenses in addition to attacks, mechanisms for monitoring misuse, mechanisms to monitor how a system learns from feedback over time, improving the efficiency and accessibility of ML).

11. **Safeguards**

Question: Does the paper describe safeguards that have been put in place for responsible release of data or models that have a high risk for misuse (e.g., pretrained language models, image generators, or scraped datasets)?

Answer: [NA]

Justification: Models trained on toy tasks. Doesn't post risk of misuse.

Guidelines:

- The answer NA means that the paper poses no such risks.
- Released models that have a high risk for misuse or dual-use should be released with necessary safeguards to allow for controlled use of the model, for example by requiring that users adhere to usage guidelines or restrictions to access the model or implementing safety filters.
- Datasets that have been scraped from the Internet could pose safety risks. The authors should describe how they avoided releasing unsafe images.
- We recognize that providing effective safeguards is challenging, and many papers do not require this, but we encourage authors to take this into account and make a best faith effort.

12. **Licenses for existing assets**

Question: Are the creators or original owners of assets (e.g., code, data, models), used in the paper, properly credited and are the license and terms of use explicitly mentioned and properly respected?

Answer: [Yes]

Justification: We provide citation and describe the license and terms of use in Section 2 for the dataset we use.

Guidelines:

- The answer NA means that the paper does not use existing assets.
- The authors should cite the original paper that produced the code package or dataset.
- The authors should state which version of the asset is used and, if possible, include a URL.
- The name of the license (e.g., CC-BY 4.0) should be included for each asset.

- For scraped data from a particular source (e.g., website), the copyright and terms of service of that source should be provided.
- If assets are released, the license, copyright information, and terms of use in the package should be provided. For popular datasets, `paperswithcode.com/datasets` has curated licenses for some datasets. Their licensing guide can help determine the license of a dataset.
- For existing datasets that are re-packaged, both the original license and the license of the derived asset (if it has changed) should be provided.
- If this information is not available online, the authors are encouraged to reach out to the asset's creators.

13. **New Assets**

    Question: Are new assets introduced in the paper well documented and is the documentation provided alongside the assets?

    Answer: [Yes]

    Justification: As of now no new assets are being released although we plan to open-source our code at a later point.

    Guidelines:

    - The answer NA means that the paper does not release new assets.
    - Researchers should communicate the details of the dataset/code/model as part of their submissions via structured templates. This includes details about training, license, limitations, etc.
    - The paper should discuss whether and how consent was obtained from people whose asset is used.
    - At submission time, remember to anonymize your assets (if applicable). You can either create an anonymized URL or include an anonymized zip file.

14. **Crowdsourcing and Research with Human Subjects**

    Question: For crowdsourcing experiments and research with human subjects, does the paper include the full text of instructions given to participants and screenshots, if applicable, as well as details about compensation (if any)?

    Answer: [NA]

    Justification: NA

    Guidelines:

    - The answer NA means that the paper does not involve crowdsourcing nor research with human subjects.
    - Including this information in the supplemental material is fine, but if the main contribution of the paper involves human subjects, then as much detail as possible should be included in the main paper.
    - According to the NeurIPS Code of Ethics, workers involved in data collection, curation, or other labor should be paid at least the minimum wage in the country of the data collector.

15. **Institutional Review Board (IRB) Approvals or Equivalent for Research with Human Subjects**

    Question: Does the paper describe potential risks incurred by study participants, whether such risks were disclosed to the subjects, and whether Institutional Review Board (IRB) approvals (or an equivalent approval/review based on the requirements of your country or institution) were obtained?

    Answer: [NA]

    Justification: NA

    Guidelines:

    - The answer NA means that the paper does not involve crowdsourcing nor research with human subjects.

- Depending on the country in which research is conducted, IRB approval (or equivalent) may be required for any human subjects research. If you obtained IRB approval, you should clearly state this in the paper.
- We recognize that the procedures for this may vary significantly between institutions and locations, and we expect authors to adhere to the NeurIPS Code of Ethics and the guidelines for their institution.
- For initial submissions, do not include any information that would break anonymity (if applicable), such as the institution conducting the review.

