# OpenReview forum: "Causal language modeling can elicit search and reasoning capabilities on logic puzzles"
_NeurIPS.cc/2024/Conference — NeurIPS 2024 poster_

### Official Review · Reviewer_Uxpn · 2024-07-10

**Soundness:** 3
**Presentation:** 4
**Contribution:** 3
**Rating:** 8
**Confidence:** 4

**Summary:**

This work attempts to solve a filtered list of Sudoku puzzles by training a transformer model with data derived from solutions produced by a mechanistic 'simple solver' (rather than a sophisticated recursive planner).  They show that the training regime transformer model can be engineered to enable the model to learn to do the task effectively.  In addition, the authors perform linear probes to show the extent of the model's internal representation of the ongoing problem solution.

**Strengths:**

This work tackles a problem that is readily understood by the public, using transformers which might be expected to struggle (compared to GOFAI solvers, which work on even harder Sudoku puzzles than tested here).

The number of experiments, looking at different aspects of the learnability of the task, and the probing of the resulting model internal states is very nicely done.

**Weaknesses:**

Selecting only those solvable by a simple solver is quite a simplification : Puzzles that require a full-backtracking approach are excluded.  This means that the results show that Transformers are capable of planning when the situation is simple, which is far from being fully capable of planning/reasoning.

Should link to dataset (https://www.kaggle.com/datasets/radcliffe/3-million-sudoku-puzzles-with-ratings).
According to the dataset description on Kaggle, 43% of the puzzles in the dataset have a difficulty of zero, meaning that it can be solved using a simple scanning technique.  The filtered dataset is 1.9M of the original 3.0MM (63%), so only 31% of the dataset being used is not amenable to the 'simple scanning technique'.  Perhaps this should be highlighted.

Should mention the 42M param GPT-2 architecture earlier in the paper than Appendix B.

Minor quibbles:

* L32: "In this work, we aim to understand how complex a reasoning task can Transformers trained with next-token prediction solve by focusing on a synthetic task: Sudoku puzzles."
  + ->   "In this work, we aim to understand how complex a reasoning task Transformers trained with next-token prediction can solve by focusing on a synthetic task: Sudoku puzzles."

* L70: "This ensures that all our puzzles are solvable in polynomial time."  Should be 'reasonable time' - the polynomial time claim is beyond what's proven.

**Questions:**

Could an analysis of model performance vs the 'rating' of the puzzle hardness (which I believe exists in the dataset) be done?  i.e. Does the model get more puzzles wrong, the objectively harder they get?

**Limitations:**

The filtering of the initial dataset should have been emphasised more : It may be that the model learns only puzzles that would be in an 'Easy Sudoku' collection.

---

> ### Author Rebuttal · Authors · 2024-08-07
>
> We thank the reviewer for their review and constructive feedback. We will make a pass over the paper and fix the typos pointed out and any other errors we find. We address your comments below.
>
> - *Selecting puzzles solvable by a solver*: We emphasize that we did this filtering on both our train and test sets to ensure that the puzzles can be solved in reasonable time. Additionally, although our solver only uses 7 strategies, many of the strategies are highly advanced (a hobby Sudoku solver might not be aware of some of them). We thus argue that our solver is a complex algorithm, albeit one which doesn’t use backtracking. We agree that this rules out puzzles which can be solved in reasonable time with backtracking and hence our results currently do not show that causal language modeling on transformers yields them with planning capabilities in complex scenarios requiring extensive backtracking.
>
> - *Difficulty rating of puzzles on Kaggle*: We wish to point out that the difficulty rating provided on Kaggle is an imperfect measure of the difficulty of the puzzle. This rating is computed as follows. To rate a puzzle, it consider a solver (different from the one we use to generate our solver-order data) which tries to iteratively make progress on a puzzle using some elimination techniques. When the solver gets stuck, it makes guesses and tries to solve the puzzle. The difficulty rating is the average number of guesses the solver had to make to solve the puzzle. Therefore, even a puzzle rated 0.5 can require complex strategies beyond simple scanning to solve them without guessing. Moreover, even for puzzles with rating 0, the elimination technique employed by the solver includes the hidden single strategy which is quite computationally intensive. That being said, we provide an analysis about how the complete puzzle accuracy changes as we increase the rating of puzzle in the attached PDF of the global response.
>
> - *“Polynomial time claim”*: To clarify the claim, by polynomial time we mean that polynomial in $n$ if the size of the puzzles is $n \times n$.  We say that all our filtered puzzles are solvable in polynomial time because if we consider the general version of Sudoku of size $n \times n$ and also consider a generalized version of the solver that employs the same set of strategies, then all the examples filtered by the solver will be solved by polynomial time in $n$ because each strategy in the solver runs in polynomial time in $n$ and none of the filtered Sudoku requires guessing a value in some empty cell. We will clarify this claim in the revised version of the paper.
>
> In addition to these comments, we have added additional ablations and error analysis in the general response above. We have also shown additional experiments in a different puzzle solving setting - solving Zebra puzzles (also known as Einstein riddles) which shares some similarity with Sudoku puzzles but has many differences. We observe a similar outcome with Zebra puzzles as well where causal language modeling on solver order data is able to teach the model to perform well on these puzzles. Please see the general response for more details on these experiments.

---

> > ### Comment · Reviewer_Uxpn · 2024-08-10
> >
> > * "Selecting puzzles solvable by a solver" : Makes sense.  But wouldn't hurt to be upfront in the paper.
> >
> > * "Difficulty rating of puzzles on Kaggle" : Good addition, makes the results stronger, by showing that success rates roughly align with difficulty (particularly since it's calculated independently)
> >
> > * "Polynomial time claim" : I understood the scaling part at L46.  By the time you get to L70, n is a fixed constant.  So "polynomial" just sounds misplaced.
> >
> > Sticking with my scores.

---

### Official Review · Reviewer_Qf5M · 2024-07-12

**Soundness:** 2
**Presentation:** 2
**Contribution:** 2
**Rating:** 5
**Confidence:** 3

**Summary:**

This paper applies causal language models (Transformers) to solve Sudoku puzzles, reporting a 94.21% accuracy rate in solving the puzzles completely. The authors claim to demonstrate that these models can develop human-like reasoning capabilities by employing insights from CoT prompting through carefully structured training data and probing analyses.
The contribution of the paper is:
1. Demonstration that causal language models can learn complex reasoning tasks like Sudoku solving through carefully structured training data.
2. Development of a novel training methodology that leverages solver-decomposed reasoning order to enhance model performance.

**Strengths:**

Novel Methodology and Application: Applying Transformers to Sudoku Solving is an interesting extension of Transformers techniques, and the use of sequences mimicking human reasoning steps is a creative training method.

**Weaknesses:**

1: Limited novelty: While applying Transformers to Sudoku is new, the underlying techniques are not innovative. The paper lacks significant theoretical or methodological advancements.

2: Narrow scope: Focusing solely on Sudoku limits the paper's impact and generalizability. It's unclear how well this approach would generalize to other reasoning tasks or more complex Sudoku variants. Testing models across various types of reasoning tasks, especially those requiring different logical structures or knowledge types (rule-based vs. rule-less puzzles), could significantly enhance the understanding of the model's generalization capabilities. [1]

3: Inadequate comparisons: A major oversight is the lack of comparisons to traditional Sudoku-solving algorithms or other AI approaches.  It's necessary to compare neural approaches with traditional algorithms to assess advancements meaningfully [1]

4: Overstated claims: The paper may overstate the model's "reasoning" capabilities. It's not yet convincing that what's described could be pattern matching rather than actual reasoning. Distinguishing between genuine reasoning and sophisticated pattern matching can be challenging. Further evidence could be demonstrated by testing the model's ability to solve puzzles it was not directly trained on, or by altering puzzle formats to see if the model can adapt its solving strategy without retraining.

5: Computational efficiency: There's insufficient discussion of the computational costs involved comparing the different order/decomposing as well as beam search appraoches.

6: Lack of error analysis: A detailed examination of where and why the model fails would provide more insight than focusing on its successes.

[1] Giadikiaroglou, P., Lymperaiou, M., Filandrianos, G., & Stamou, G. (2024). Puzzle Solving using Reasoning of Large Language Models: A Survey.

**Questions:**

1. How does this approach compare speed and efficiency to traditional Sudoku-solving algorithms?
2. What evidence supports the claim that the model is genuinely "reasoning" rather than pattern matching?
3. How well does this method generalize to other types of puzzles or reasoning tasks?
4. What are the computational efficiency for different tested methods presented?
5. Can you provide a more detailed error analysis? what are standard failure modes?
6. Have you investigated whether the model can explain its reasoning process, similar to how humans might describe their Sudoku-solving steps？

**Limitations:**

Yes

---

> ### Author Rebuttal · Authors · 2024-08-07
>
> We thank the reviewer for their review and constructive feedback. We address your comments below.
>
> - *Lacks theoretical or methodological advancements*: We reiterate that our main contribution is not proposing a new theory or a new method to train LLMs. Rather it is an advance in our understanding of the precise capabilities and limitations of causal language modeling. Our work is similar to works such as [1], [2], [3] which show that Transformers can learn useful world models.
> That being said, we posit that our insight on showing the right style of data to train models on for reasoning puzzles might be helpful in developing better pre-training data collection strategies for general LLMs.
> Our contribution is novel because on general-purpose LLMs such as GPT-4, there is much evidence [5], [6], [7] that these models do not do well at challenging reasoning/planning tasks. Hence, apriori, it is not obvious that Sudoku puzzles can be solved by causal LM alone.
>
> - *Narrow scope - evaluations on other reasoning puzzles/variants*:
>   - Firstly, we note that Sudoku is a challenging task where prior work had not shown success of the causal LM approach (see point in general response about performance of frontier LLMs). While Sudoku is a specific puzzle setting, this enables us to perform a controlled study where we can quantify the exact amount of generalization we observe.Taking the reviewer’s feedback we added experiments with a second puzzle type: Zebra puzzles, where we observed similar results. Please refer to the general response and the attached PDF for more details. We will also add these to the paper.
>   - In addition, we believe that Sudokus involve many sub-reasoning tasks involving deductive logic, mapping to abstract concepts etc and are representative of many challenging deterministic puzzles. Hence, we believe, our main message is robust to changes to the puzzle such as the logical dependency structure while keeping the required components of logical inference, search, and planning the same.
>   - Rule-less puzzles require world knowledge to solve and complicate a principled and controlled study of the type we conducted. It is challenging to generate CoT style data for such puzzles on a large scale. We believe this is a very interesting direction for future research.
>   - We also like the idea of stochastic puzzles. It would be interesting to see how causal LM performs on them but we believe this is outside the scope of the current paper.
>
> - *Comparison to traditional Sudoku solvers and AI approaches*: Please see the corresponding point in the general response above.
>
> - *Overstated claims of reasoning vs pattern matching*: We disagree with this comment. Human reasoning also involves a great deal of pattern matching. However, this happens in an abstract concept space rather than the raw input space. For instance, when presented with a novel coding problem, one might perform an abstract pattern matching to understand whether dynamic programming or divide and conquer are applicable. A lot of frontier research also involves pattern matching. Hence, we argue that learning to pattern match in an abstract concept space consistently over a number of steps is reasoning. While we agree that our setting focuses on a single puzzle type, the model learns to solve unseen puzzles; it learns abstract “strategies” during training and applies them for a test puzzle. This is pattern matching in the abstract concept space. Moreover, some of these abstract strategies can involve O(n^3) computation to find the right square to apply the strategy to. The model is learning to do this search! In addition, it learns to do this consistently for over 50 steps. Hence, our claim of the model learning to reason.
>
> - *Computational Efficiency*: The computational cost of the different orders of the data in our paper is the same. In particular, giving only the value filled in each cell in a step by step manner is more efficient than giving an entire search trace as is done in [4].
> Beam search only adds a constant K^2 factor to the decoding process. While further improving the computational efficiency of LLMs is an important area of research, that is not our primary focus and we leave it for future work. We will add a detailed discussion on the computational efficiency aspects in the paper.
>
> - *Error analysis*: Thanks for the feedback. We have a preliminary error analysis in the paper. Please refer to the general response for details on this. We will be adding a more extended error analysis in the paper.
>
> - *Explaining model’s reasoning process*: Our toy model can’t generate English explanations. However, our probing study shows that the model implicitly keeps track of candidate sets (the set of possible values in a given cell). This is a commonly used technique by humans. In addition, we have done an edit distance analysis to study how much the choice of cell locations to decode varies between the model vs the solver. We find that there is a great deal of alignment between these two orders thus providing additional evidence of a human-like reasoning process.
>
> [1] Emergent world representations: Exploring a sequence model trained on a synthetic task. Li et al.
>
> [2] Physics of language models: Part 1, context-free grammar. Allen-Zhu et al.
>
> [3] Physics of Language Models: Part 2.1, Grade-School Math and the Hidden Reasoning Process. Allen-Zhu et al.
>
> [4] Beyond a*: Better planning with transformers via search dynamics bootstrapping. Lehnert et al.
>
> [5] [1] Large language models still can’t plan (a benchmark for llms on planning and reasoning about change). Valmeekam et al.
>
> [6] Travelplanner: A benchmark for real-world planning with language agents. Xie et al.
>
> [7]  Limits of transformers on compositionality. Dziri et al.

---

> > ### Comment · Reviewer_Qf5M · 2024-08-11
> > **response to the rebuttal**
> >
> > Thanks for the detailed response especially on the error analysis and the additional study on the zebra puzzle; I still hope to see more experimental results on comparison with other solvers and models to justify its effetiveness, but I am willing to increase my rating for the current draft.

---

### Official Review · Reviewer_xEc8 · 2024-07-14

**Soundness:** 3
**Presentation:** 3
**Contribution:** 2
**Rating:** 5
**Confidence:** 4

**Summary:**

* This work presents a study of solving sudoku puzzles via causal language modeling.
* Given the sequence of filled places and their values in sudoku, the model must output the series of empty cell positions and the values that correspond to them.
* They study how the model performs with various input representations of a sudoku puzzle. (considering sudoku puzzle as a matrix)
    * Fixed cell order (from top-left to bottom-right)
    * Random cell order
    * Chain-of-thought prompting (using a solver to provide the method to solve the sudoku)
* Through experiments, the authors have demonstrated that appropriate training data that breaks down the problem into smaller components is essential for the model to solve the puzzle correctly.
    * The model's performance improved with CoT prompting. It's even enhanced with the use of position hints and beam search.

**Strengths:**

* The problem definition, model description, and experimental setup are presented clearly, making the paper accessible and informative.
* Introduced a sudoku puzzle dataset with steps to solve the puzzles (1.9M puzzles).
* Probing analysis for tracking the candidate set of cell,

**Weaknesses:**

1. No SoTA models are evaluated on the sudoku puzzle data.
2. A full ablation analysis is not included. This is to understand better how different settings (CoT, beam search, puzzle complexities) affected the model performance and where the model is struggling. Only improvements in accuracies are mentioned in the paper.

**Questions:**

1. How do models like GPT-4, Gemini-1.5, and Llama-3 perform on these sudoku puzzles? (with and without CoT)
2. Are the plots in Figure 3 in the CoT+Beam search setting?

---

> ### Author Rebuttal · Authors · 2024-08-07
>
> We thank the reviewer for their review and constructive feedback. We will make a pass over the paper and fix the typos pointed out and any other errors we find. We address your specific comments and questions below.
> - *No SoTA models are evaluated on the sudoku puzzle data*: We reiterate that our main focus is not to compete with SoTA approaches to Sudoku solving. We are studying the capabilities and limitations of causal language modeling (which is accepted as the dominant approach to train language models today) in a controlled setting and not looking to compare with other approaches. Nonetheless in our general response above, we present a comparison with some other approaches to solve Sudoku puzzles and will add this discussion to the paper. Note that in our approach we don’t handcraft any parts of the architecture for Sudoku puzzles.
> - *How do models like GPT-4, Gemini-1.5 and Llama-3 perform on these Sudoku puzzles (with and without CoT)?*: Frontier LLMs like GPT-4 and Gemini-1.5 are expected to perform very poorly on a challenging reasoning task such as Sudoku. The poor performance of these models on planning/reasoning tasks has been reported in [1], [2], [3] and others. For completeness, we performed a small-scale study on Gemini-1.5 and GPT-4o where we queried them with 3000 Sudoku puzzles each  (in a 4-shot with CoT manner) and we analyzed how well they can solve the puzzle. Overall, we found that they got 0% of the puzzles completely right and their accuracy on a per-cell basis was around 8-10% which is close to random guessing. We will include these results in the paper.
> - *A full ablation analysis is not included in the paper*: Thank you for this feedback. We will be extending the number of ablation studies we have in our paper. We wish to point out that we did include some ablations such as experiments with CoT data vs random order data vs fixed order data. In addition we have also included ablations for different beam search widths. In addition, we will have now performed ablations with respect to puzzle difficulties. The results are described in the general response.
> - *Are the plots in Figure 3 in the CoT+Beam search setting?*: We apologize for the confusion. These plots are just CoT training without beam search. We will clarify this in the figure’s caption.
>
>
> In addition to these comments, we have added additional ablations and highlighted some error analysis into where the model struggles in the general response above. We have also shown additional experiments in a different puzzle solving setting - solving Zebra puzzles (also known as Einstein riddles) which shares some similarity with Sudoku puzzles but has many differences. We observe a similar outcome with Zebra puzzles as well where causal language modeling on solver order data is able to teach the model to perform well on these puzzles. Please see the general response for more details on these experiments.
>
> [1] Large language models still can’t plan (a benchmark for llms on planning and reasoning about change). Valmeekam et al. 2022.
>
> [2] Travelplanner: A benchmark for real-world planning with language agents. Xie et al. 2024
>
> [3]  Limits of transformers on compositionality. Dziri et al. 2024.

---

> > ### Comment · Reviewer_xEc8 · 2024-08-12
> >
> > My concerns are addressed. Thank you for your response.

---

> ### Author Response · Authors · 2024-08-12
>
> Thank you for your response! Let us know if there are any additional comments/concerns that we can answer.
>
> If we have addressed your concerns, can you reconsider the score? Thank you.

---

### Official Review · Reviewer_W1Mz · 2024-07-15

**Soundness:** 2
**Presentation:** 2
**Contribution:** 3
**Rating:** 7
**Confidence:** 3

**Summary:**

This paper assesses causal language models', particularly transformer decoders', abilities to solve Sudoku puzzles. The authors encode Sudoku puzzles into sequences, representing each cell as a (row, column, value) triple, and train a model from scratch on 1.8M puzzles. They then evaluate the trained model on a 0.1M holdout test set. Results indicate that when unfilled cells in the training data are arranged in an easy-to-hard order (based on a solver's results), the model can solve Sudoku puzzles with a 94.21% full-puzzle solving rate. The authors also use linear probing to show that the model's activations contain information about possible values for any given cell. The paper concludes that causal language models may exhibit complex reasoning capabilities when trained on data that informs appropriate reasoning steps, without requiring techniques like Chain-of-Thought or external reasoning engines.

**Strengths:**

- The selected task and settings are suitable for studying the reasoning and planning capabilities of language models;
- The paper presents strong results that causal LMs can solve the Sudoku puzzle by training with appropriate data, without the need of techniques such as using CoT, search or external solvers.
- The results indicate that causal LMs may be able to perform search and planning internally, which seems novel and insightful for further research.
- The writing is easy to follow.

**Weaknesses:**

- The probing study's methodology is somewhat questionable. It merely compares the top-k predictions for each cell against the ground truth candidate set. This approach may not accurately be termed "probing" as it doesn't examine intermediate representations. Furthermore, this study might not conclusively demonstrate that the language model internally tracks candidate sets, given that the model is explicitly prompted to predict the cell. A more effective approach could involve probing potential values of one cell while prompting the model to predict another. Positive results from such a method could more convincingly show that the model can internally reason about other cells relevant to solving the current one.
- I’m not sure about what the takeaway of Sec. 3.5 is.
- It seems like the paper used the wrong template.

**Questions:**

- Showing some example data in the appendix can be helpful.
- When trained with random order data, do you include tokens for cell locations for the loss calculations? Additionally, how do you ensure that the model predicts every cell during testing?
- I assume the input context always includes previously predicted cells and values, is this correct?

**Limitations:**

The authors have properly adequately limitations of their work in the paper.

---

> ### Author Rebuttal · Authors · 2024-08-07
>
> We thank the reviewer for their review and constructive feedback. We will make a pass over the paper and fix the typos pointed out and any other errors we find. We address your comments below.
>
> - *Clarifying our probing methodology*: We agree that the more common way to perform a probing analysis involves learning a linear/non-linear model which takes in the embedding and outputs a label indicating a concept. However, we use probing in more general sense to refer to understand some of the innerworkings of the model. In addition, we want to clarify some confusion about how we do our probing analysis. We DO NOT only evaluate the top-k candidates on the cell the model chooses to predict. Although during its natural course of decoding the model might wish to decode cell location A, we force it (by conditioning) to decode at every other location and evaluate the top-k candidates.
>
> - *Probing potential values of one cell while predicting another*: We had considered this approach but felt it was harder to get it working for the following reasons. It is unlikely that the same probe will work across different cells (since the embedding conditioned for predicting a particular cell might suppress information about other cells). This might require us to train separate probes for the values of each cell for which the amount of data available becomes sparse. We did try an alternative approach instead where we take the whole sequence of embeddings the model has produced as input to the probe. However, this concatenated embedding becomes a ~25000 dimensional vector which is very high dimensional and makes it hard to train the probe.
>
> - *Takeaway of Section 3.5*: This section shows the additional gains we get when we use beam search decoding instead of greedy decoding. Beam search decoding with beam width K, maintains K `plausible’ candidates to continue decoding at all times. Note that we prevent a combinatorial explosion with the decoding length by always truncating the list to what we believe are the K most plausible prefixes so far. Using this approach, we show that we are able to further boost the complete puzzle solving accuracy. The main takeaway from this section is that, even in situations where the model’s most likely next token is incorrect, the correct token is in the top-K most likely tokens.
>
> - *Wrong Template*: We are unsure of what the reviewer meant by the wrong template. We apologize if a formatting error slipped past us. Please let us know what specifically you are referring to and we will fix it.
>
> We answer the reviewer’s technical questions below:
>
> - *Showing example data in the appendix*: We thank the reviewer for this feedback. Indeed, we agree that this would help explain the challenges in our work better and we will add examples to this end in the Appendix.
>
> - *When training with random data, do you include cell location tokens for loss calculations?*: We have tried both including it and excluding it. The results are similar in both settings.
>
> - *Additionally, how do you ensure that the model predicts every cell during testing?*: We deliberately don’t explicitly ensure this. The model is supposed to learn the basic rules of Sudoku as well purely from data. How well they are able to adhere to the rules during inference is a measure of their generalization ability.
>
> - *I assume that the input context always includes previously predicted cells and values*: Yes this is correct. Note that the input puzzle context length can vary as different puzzles have a different number of filled cells to begin with.
>
> In addition to these comments, we have added additional ablations and a detailed error analysis into where the model struggles in the general response above. We have also shown additional experiments in a different puzzle solving setting - solving Zebra puzzles (also known as Einstein riddles) which shares some similarity with Sudoku puzzles but has many differences. We observe a similar outcome with Zebra puzzles as well where causal language modeling on solver order data is able to teach the model to perform well on these puzzles. Please see the general response for more details on these experiments.

---

> ### Comment · Reviewer_W1Mz · 2024-08-12
>
> Thank you for your response and the additional results. My concerns are mostly addressed. I like the additional results on the Zebra puzzles and I encourage the authors to include them in the next revision.
> - Regarding the probing methodology: I'm still unsure this is the best way to perform probing. However, I agree that the current setup can provide evidence that the model is solving the puzzle in a way that resembles that of a logical solver.
> - Additional notes on the difficulty measurement: in Figure 1 of the general response, I notice that the model performs almost perfectly on the easiest puzzles and performs worse and worse as the difficulty increases. Although this behavior is expected, I think it's worth discussing the difficulty distribution of the training and test data. Especially, how will the training data mixing affect the generalization performance to other difficulties? If the model truly learns the reasoning strategy, should we expect it to generalize from hard to simple questions? I feel like there are many interesting points that can be explored here.
>
> Regardless, I'm happy to increase the rating of the current version.

---

### Author Rebuttal · Authors · 2024-08-07

We thank all the reviewers for their careful reviews and constructive feedback. We first address comments raised by multiple reviewers.

- *Comparison to traditional solvers and AI approaches*: We reiterate that our main focus is not to propose a new approach to solve the sudoku puzzle. We study the capabilities and limitations of causal language modeling (which is the dominant approach to train LLMs today) in a controlled setting. Note that in our approach we *don’t handcraft any parts of the architecture or the loss function* for Sudoku puzzles. Nonetheless we present here a comparison with some other approaches and will include a discussion in the revised draft:
  - Combinatorial (Traditional) solvers: SoTA combinatorial solvers can solve a much larger fraction of Sudoku puzzles than our method. In fact, we filter to train and test only on those puzzles solvable by one such combinatorial solver. So our performance on a more general test set will be worse than such solvers. But this is an unfair comparison and not the point of the paper as these solvers are handcrafted with human intellect.
  - Frontier LLMs: Frontier LLMs like GPT-4 and Gemini-1.5 are expected to perform very poorly on a challenging search and reasoning task such as Sudoku. The poor performance of these models on planning/reasoning tasks has been reported in [1,2,3]. Also see point 2 below.
  - “Large Language Model Guided Tree-of-Thought” by Jieyi Long: This paper studies tree-of-thought prompting of LLMs and gets it to work for 5x5 Sudoku puzzles. This is an expensive prompting scheme already for 5x5 puzzles and these puzzles are much easier than 9x9 puzzles.
  - “Recurrent Relational Networks” by Palm et al.: This paper handcrafts the recurrent network to match the puzzle structure (and obey the constraints) and performs multiple rounds of message passing between cells of the sudoku puzzle to arrive at a solution. We evaluate our trained model (trained using causal language modeling) on the test dataset proposed in Palm et al. and we observe a comparable performance without handcrafting the network or loss function (see attached PDF for the result).
  - “Solving Sudoku with neural networks” by Akin-David et al.: They study how well handcrafted CNNs or LSTMs work for solving Sudoku puzzles and observe worse results than us.

- *Experiments with Frontier LLMs*: We performed a small-scale study on Gemini-1.5 and GPT-4o where we queried them with 3000 Sudoku puzzles each  (in a 4-shot with CoT manner). Overall, we found that they got **0% of the puzzles completely** right and their accuracy on a per cell basis was around **8-11% (close to random guessing)**. We will add these results to the paper.

- *Error analysis*: We have some preliminary error analysis in the paper. Figure 3 (Appendix) shows where during decoding the model makes the first error. It tends to make it more often in the first 10-15 steps of decoding than later. We have added a breakdown of accuracy vs puzzle difficulty in the attached PDF. We also found some examples where there exists some easy-to-decode cells but the model fails to find such cells and therefore tries to decode a harder cell and ends up making a mistake. We have added one such example in the attached PDF. This adds to the evidence (as already mentioned in the paper) that when we provide hints about easy to decode positions to the model, it predicts the value of the cell correctly ~100% of the time. This indicates that the model is struggling to search for the positions that are easy to decode and the model selects positions which can’t be solved with the current information sometimes.

- *Additional experiments with Zebra puzzles*: To extend our analysis beyond sudoku puzzles as requested by the reviewers, we conduct our experiments for the Zebra puzzle (also known as Einstein's Puzzle) [4].
  - *Background*: The Zebra puzzle is characterized by no. of entities and no. of attributes for each entity. e.g., each house in the original Einstein’s puzzle [4] is an entity and color, nationality, drink, smoke and pet are attributes associated with each house. In the puzzle, clues about relationships between entities and attribute values are given and the task is to figure out values for all attributes and for all the entities. Please see [4, 5] for some example puzzles. Observe that each clue can be abstracted out to a relationship between value of attributes and entities. e.g., “The Englishman lives in the red house” clue says that house (entity) which has color attribute = red also has nationality attribute = Englishman.
  - *Experimental details*: To generate a zebra puzzle, we start with a random permutation of entities, attributes and values of each attribute and then we keep adding clues until the solver can solve the zebra puzzle without guessing/backtracking. To obtain a solution for a puzzle, the solver keeps track of possible values of each attribute for each entity and tries to narrow down them using the clues. The solver tries to make progress with clues with easier reasoning steps first and if it isn’t able to make progress then it tries clues with hard reasoning steps. Thus, the solver will fills the value of attributes which are easier to decide before going to the harder ones. Similar to the sudoku puzzle, we use the order in which attribute values are filled by the solver to train a transformer model of the same size using Causal LM (using the same hyperparameter as in Sudoku puzzle). We report our results in the attached pdf. The trained model solves **92% of the puzzles completely** and for **96% of the attributes**, the model predicts the correct value.

[1] Large language models still can’t plan. Valmeekam et al. 2022.

[2] Travelplanner: A benchmark for real-world planning with language agents. Xie et al. 2024

[3]  Limits of transformers on compositionality. Dziri et al. 2024.

[4] Zebra Puzzle. Wikipedia page.

[5] Zebra puzzle on Brainzilla webpage.

---

### Decision · Program_Chairs · 2024-09-25

**Decision:**

Accept (poster)

**Comment:**

the paper probes the order in which reasoning puzzles (sudoku) get filled and its impact on performance. Paper has good insights and clear conclusions which are liked by reviewers. All of them lean positive -- so seems like a clear accept.

Authors should incorporate all discussion points in the revised version.